# SynBrain: Enhancing Visual-to-fMRI Synthesis via Probabilistic Representation Learning

Weijian Mai[1,2]*,   Jiamin Wu[1,3]†,   Yu Zhu[1],   Zhouheng Yao[1],   Dongzhan Zhou[1],
Andrew F. Luo[2],   Qihao Zheng[1],   Wanli Ouyang[1,3],   Chunfeng Song[1,4]

[1]Shanghai Artificial Intelligence Laboratory    [2]The University of Hong Kong
[3]The Chinese University of Hong Kong    [4]Shanghai Innovation Institute

## Abstract

Deciphering how visual stimuli are transformed into cortical responses is a fundamental challenge in computational neuroscience. This visual-to-neural mapping is inherently a one-to-many relationship, as identical visual inputs reliably evoke variable hemodynamic responses across trials, contexts, and subjects. However, existing deterministic methods struggle to simultaneously model this biological variability while capturing the underlying functional consistency that encodes stimulus information. To address these limitations, we propose *SynBrain*, a generative framework that simulates the transformation from visual semantics to neural responses in a probabilistic and biologically interpretable manner. SynBrain introduces two key components: (i) **BrainVAE** models neural representations as continuous probability distributions via *probabilistic learning* while maintaining functional consistency through visual semantic constraints; (ii) A **Semantic-to-Neural Mapper** acts as a semantic transmission pathway, projecting visual semantics into the neural response manifold to facilitate high-fidelity fMRI synthesis. Experimental results demonstrate that SynBrain surpasses state-of-the-art methods in subject-specific visual-to-fMRI encoding performance. Furthermore, SynBrain adapts efficiently to new subjects with few-shot data and synthesizes high-quality fMRI signals that are effective in improving data-limited fMRI-to-image decoding performance. Beyond that, SynBrain reveals functional consistency across trials and subjects, with synthesized signals capturing interpretable patterns shaped by biological neural variability. Our code is available at https://github.com/MichaelMaiii/SynBrain.

## 1 Introduction

Understanding how the human brain transforms visual stimuli into structured patterns of neural activity remains one of the fundamental challenges in computational neuroscience [37, 24, 18, 54]. This task, commonly referred to as *brain visual encoding*, seeks to model the functional mapping from external visual perception to spatially distributed neural responses across the cortex, uncovering how high-level visual semantics are represented in neural populations [31, 32, 4]. In recent years, functional magnetic resonance imaging (fMRI) has emerged as a dominant modality for brain encoding. As a noninvasive neuroimaging technique, fMRI measures blood-oxygen-level-dependent signals as indirect proxies for neuronal activity with high spatial resolution [38]. By translating visual inputs into fMRI-measured neural patterns, brain encoding models not only advance our understanding of human visual perception but also lay the groundwork for applications in neural decoding [35, 8, 47, 49, 28, 34, 60], cognitive modeling [63, 66], and brain–machine interfaces [13, 26].

---

*This work was done during his internship at Shanghai Artificial Intelligence Laboratory.
†Corresponding Author (wujiamin1@pjlab.org.cn)

39th Conference on Neural Information Processing Systems (NeurIPS 2025).

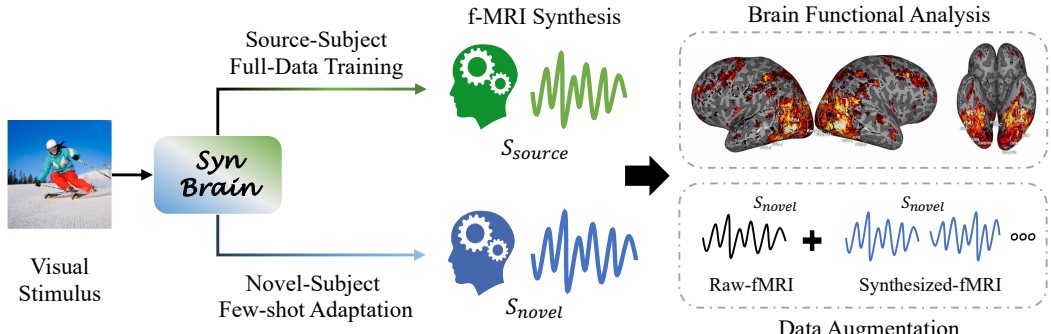

Figure 1: Overview of SynBrain for subject-adaptive visual-to-fMRI synthesis and downstream decoding applications. SynBrain is trained on full fMRI recordings from a source subject and adapted to novel subjects using limited data. It generates semantically consistent neural responses that support brain functional analysis and enhance downstream decoding through synthetic data augmentation.

Recent advances in brain encoding have predominantly adopted either **regression-based** or **deterministic generative** strategies to map visual stimuli or their semantic representations to corresponding fMRI recordings. Regression models typically learn a deterministic function that directly predicts voxel-level brain activity from visual stimuli using linear regression or deep neural networks [16, 61, 1, 33, 26, 5, 3, 32]. Differently, generative brain encoding methods such as MindSimulator [4] reformulate brain encoding as a visual-to-fMRI synthesis process conditioned on visual input. MindSimulator uses a deterministic AutoEncoder to reconstruct brain activity from semantic features, aligning fMRI latent space to visual stimuli via diffusion-based semantic-to-latent mapping.

However, such deterministic strategies in existing methods fail to model the inherent one-to-many mapping nature of neural signals. Specifically, existing large-scale neuroimaging studies [23, 2] indicate that repeated presentations of the same visual stimulus can elicit notably different fMRI responses across trials and subjects. This highlights a fundamental characteristic of brain activity: *the mapping from visual stimuli to neural responses is inherently one-to-many*, influenced by trial-level noise, brain attentional fluctuations, and inter-individual variability. Given this neural property, existing methods face three key limitations. **(1) Deterministic neural modeling**: previous deterministic models struggle with the one-to-many nature of brain responses, as they produce a unique latent representation per input and often collapse diverse neural patterns into a non-informative averaged response that lacks semantic or physiological validity. Although MindSimulator's diffusion-based latent mapping attempts to introduce variability through stochastic sampling, its core fMRI synthesis module relies on a deterministic AutoEncoder without probabilistic modeling. **(2) Lack of functionally-consistent variability:** current approaches fail to model neural responses that are simultaneously variable in their specific patterns yet consistent in their functional encoding of stimulus information. (3) **Limited utility as synthetic training data:** the lack of cross-subject transferability restricts their applicability as augmentation sources for neural decoding in data-limited scenarios. These challenges motivate a key research question: *how can we effectively model neural response distributions to visual stimuli with functional consistency?*

To address these limitations, we propose *SynBrain*, a generative framework that models fMRI responses as a **semantic-conditioned, continuous probability distribution**. SynBrain is designed to simulate the transformation from visual semantics to brain activity in a manner that is probabilistic and biologically interpretable. At the core of SynBrain is **BrainVAE**, a well-designed variational model that incorporates convolutional feature extractors and attention modules to mitigate unstable training dynamics observed in MLP-based variational architectures. Unlike deterministic autoencoders, BrainVAE learns a **probabilistic latent space** of fMRI responses explicitly conditioned on high-level visual semantics, allowing it to generate diverse fMRI signals that reflect neural variability while preserving functional consistency. Moreover, we incorporate **Semantic-to-Neural (S2N) Mapper**, a lightweight Transformer module that establishes a point-to-distribution mapping from fixed CLIP semantic embeddings to the probabilistic latent space of BrainVAE. S2N Mapper achieves one-step semantic-to-neural mapping for stable and semantically coherent fMRI synthesis, avoiding the distribution mismatch problems observed in previous multi-step diffusion-based methods [4].

Through this architecture, SynBrain facilitates a biologically grounded probabilistic framework for visual-to-fMRI synthesis, effectively balancing voxel-level neural fidelity with semantic consistency.

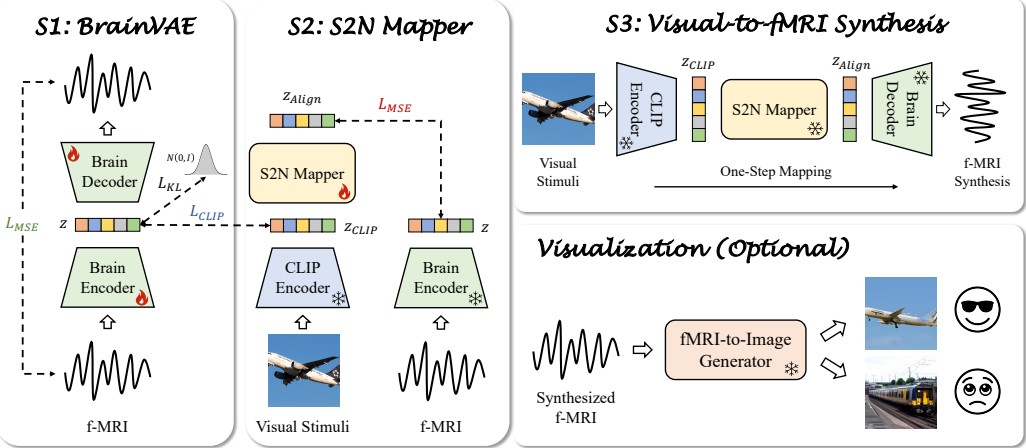

Figure 2: Overview of the SynBrain framework. **Stage 1:** BrainVAE models the probabilistic distribution of fMRI responses conditioned on CLIP visual embeddings $z_{CLIP}$; **Stage 2:** S2N Mapper learns to map $z_{CLIP}$ into the latent space of BrainVAE; **Stage 3:** At inference, the frozen S2N Mapper performs a one-step mapping from $z_{\text{CLIP}}$ to the BrainVAE latent space for visual-to-fMRI synthesis. Synthesized fMRI could be further visualized via a pretrained fMRI-to-image generator.

Specifically, its advantages are reflected in two aspects: 1) **Superior subject-specific encoding performance**: through probabilistic modeling of trial-level functional consistency, SynBrain significantly outperforms prior methods in subject-specific visual-to-fMRI synthesis. 2) **Effective few-shot adaptation to novel subjects and utility as synthetic training data**: SynBrain demonstrates strong generalization to unseen subjects in data-limited settings. By disentangling subject-specific variability from semantic representations, it transfers the synthesis capability for new subjects with minimal supervision while maintaining functional alignment. Furthermore, the high-fidelity synthesized signals can serve as effective data augmentations for downstream fMRI-to-image decoding under data-limited conditions, as shown in Figure 1. Our contributions are as follows:

- We propose SynBrain, a generative framework for visual-to-fMRI synthesis that models neural responses as semantic-conditioned probability distributions. By integrating probabilistic learning, SynBrain captures the one-to-many nature of neural responses to better reflect biological variability and synthesize semantically consistent neural responses.

- SynBrain advances visual-to-fMRI synthesis by achieving superior subject-specific and few-shot novel-subject adaptation performance. Moreover, its high-quality synthesized signals prove effective as data augmentations for improving fMRI-to-image decoding in data-scarcity scenarios.

- SynBrain reveals cross-trial and cross-subject functional consistency through brain functional analysis, demonstrating that the synthesized signals preserve interpretable cortical patterns reflecting underlying biological neural variability.

## 2 Methodology

The fundamental goal of visual-to-fMRI synthesis is to generate brain activity patterns that faithfully reflect the semantics of visual stimuli while capturing the inherent variability observed in biological neural responses. However, most existing approaches rely on deterministic mappings from vision to fMRI, producing a single fixed output for each input. These methods fail to model the ***one-to-many*** nature of brain responses, which is influenced by trial-level noise, attentional fluctuation, and individual-specific neural patterns.

To address this issue, we propose SynBrain, a generative framework that synthesizes fMRI responses from visual stimuli by modeling the semantic-conditioned distribution of neural activity. SynBrain consists of two key components: 1) **BrainVAE**, a variational backbone for semantic-guided variational

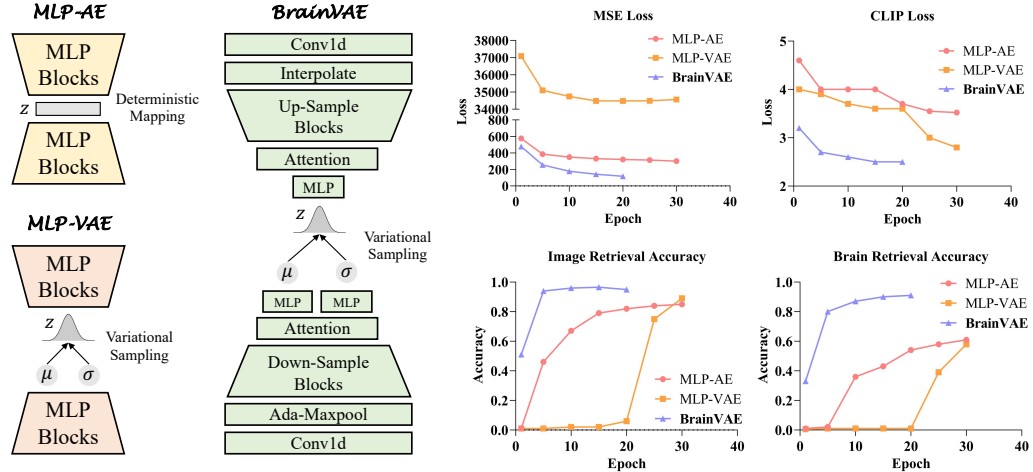

Figure 3: Architecture and performance comparison of MLP-based baselines and our proposed BrainVAE. **Left:** Architecture comparisons; **Right:** Validation performance comparisons.

modeling of neural response distributions, and 2) **S2N Mapper**, a semantic-to-neural projection module. The overall framework is trained in two stages and enables visual-to-fMRI synthesis during inference with an optional fMRI-to-image visualization procedure using the MindEye2 [49] generator, as shown in Figure 2.

## 2.1 BrainVAE

Human brain responses to visual stimuli are inherently variable across trials and subjects, yet preserve consistent semantic structure [21, 24]. Traditional encoding models often treat this mapping as deterministic, ignoring both within-subject and inter-subject variability. To address this, BrainVAE models fMRI signals as samples from a conditionally structured latent distribution, conditioned on the high-level semantic embedding derived from the visual input.

**Motivation and Architecture Comparison.** MindSimulator [4] has employed MLP-based autoencoders (**MLP-AE**) to map high-dimensional fMRI signals into a deterministic latent space. To explore the benefits of variational modeling, we first incorporated a variational formulation into this architecture, forming a baseline termed **MLP-VAE**. However, we found that MLP-VAE suffers from unstable training, with divergence in MSE loss (Figure 3). This might be attributed to MLP's lack of spatial inductive bias and token-wise independence. To tackle this issue, we propose **BrainVAE**, a well-designed variational architecture for stable and effective fMRI representation learning. As shown in Figure 3-Left, BrainVAE integrates convolutional layers to extract local voxel features and attention layers to capture long-range inter-voxel dependencies, leading to a smoother and more expressive latent space for enhanced fMRI synthesis. Figure 3-Right demonstrates that BrainVAE empirically possesses two clear advantages in fMRI signal synthesis: 1) **Improved voxel-level reconstruction fidelity and faster convergence** with lower validation MSE within fewer epochs; 2) **Stronger semantic expressiveness** captured in underlying neural patterns indicated by lower validation CLIP loss and higher cross-modal retrieval accuracies.

**BrainVAE Pipeline.** BrainVAE consists of an encoder-decoder architecture to reconstruct fMRI signals through a variational latent distribution. Given an fMRI input $y_{\text{fMRI}} \in \mathbb{R}^{1 \times n}$, the encoder outputs a posterior distribution $q(z|y)$ parameterized by mean $\mu$ and log-variance $\log \sigma^2$, defining a latent Gaussian distribution $z \sim \mathcal{N}(\mu, \sigma^2)$. A latent vector $z$ is sampled using the reparameterization trick and passed to the decoder to produce the reconstruction $\hat{y}_{\text{fMRI}} = \mathcal{D}(z)$. The design helps capture meaningful variations in brain responses while preserving structural consistency.

**Training Objectives.** To ensure faithful reconstruction and a probabilistic, semantically aligned latent space, BrainVAE is optimized with a composite objective consisting of three loss terms.

**1) Reconstruction Loss** $\mathcal{L}_{\mathrm{MSE}}$: This loss measures the voxel-wise mean squared error between the reconstructed fMRI signal $\hat{y}_{\mathrm{fMRI}}$ and the ground-truth input $y_{\mathrm{fMRI}}$:

$$\mathcal{L}_{\mathrm{MSE}} = \|\mathcal{D}(z) - y_{\mathrm{fMRI}}\|_2^2. \tag{1}$$

It enforces fidelity at the voxel level, ensuring that the decoder can reproduce detailed neural activity patterns from the latent representation.

**2) KL Divergence Loss** $\mathcal{L}_{\mathrm{KL}}$: The KL term regularizes the learned posterior $q(z \mid y_{\mathrm{fMRI}})$ to be close to a standard normal prior $\mathcal{N}(0, I)$:

$$\mathcal{L}_{\mathrm{KL}} = D_{\mathrm{KL}}(q(z \mid y_{\mathrm{fMRI}}) \parallel \mathcal{N}(0, I)). \tag{2}$$

This term encourages smoothness in the latent space, ensuring that similar fMRI inputs map to nearby latent codes and that sampling from the prior leads to plausible reconstructions.

**3) Contrastive Loss** $\mathcal{L}_{\mathrm{CLIP}}$: We adopt a contrastive learning objective, termed as **SoftCLIP** loss [15], to semantically ground the fMRI latent space and enhance cross-modal alignment. Given an input image $y$, we extract its visual representation $z_{\mathrm{CLIP}} = \mathcal{V}(y) \in \mathbb{R}^{m \times d}$ through a frozen CLIP visual encoder $\mathcal{V}(\cdot)$, where $m$ is the number of visual tokens and $d$ is the feature dimension. Concurrently, the fMRI signal $y_{\mathrm{fMRI}}$ is encoded via our brain encoder $\mathcal{E}(\cdot)$ to produce latent features $z = \mathcal{E}(y_{\mathrm{fMRI}}) \in \mathbb{R}^{m \times d}$. Alignment is achieved by minimizing:

$$\mathcal{L}_{\mathrm{CLIP}} = \mathrm{SoftCLIP}(z, z_{CLIP}). \tag{3}$$

This alignment encourages the latent space to encode semantic information consistent with the visual stimulus, enabling more controllable and meaningful fMRI generation. The final training loss is defined as a weighted sum of the above components:

$$\mathcal{L}_{\mathrm{BrainVAE}} = \mathcal{L}_{\mathrm{MSE}} + \lambda_{\mathrm{KL}}\mathcal{L}_{\mathrm{KL}} + \lambda_{\mathrm{CLIP}}\mathcal{L}_{\mathrm{CLIP}}. \tag{4}$$

Here we set $\lambda_{\mathrm{KL}} = 0.001$ to *softly regularize* the latent distribution while preserving reconstruction quality, and $\lambda_{\mathrm{CLIP}} = 1000$ to ensure fast convergence of semantic alignment within a few epochs.

Through this design, BrainVAE models a latent distribution of neural responses conditioned on visual stimuli, capturing trial-level variability and consistent semantic patterns. This formulation provides a probabilistic and semantically aligned space for visual-to-fMRI synthesis.

## 2.2 S2N Mapper

Contrastive learning inherently only encourages the alignment of fMRI embeddings with the vector direction of associated visual embeddings, which may lead to inconsistent embeddings [44]. Hence, we propose the Semantic-to-Neural (S2N) Mapper to improve cross-modal alignment by mapping visual embeddings directly into the fMRI latent space, forming a point-to-distribution alignment.

S2N Mapper comprises a typical Transformer [55] module with stacked multi-head self-attention layers and token-wise feedforward networks. Given visual embedding $z_{\mathrm{CLIP}} \in \mathbb{R}^{m \times d}$, the S2N Mapper implements a non-linear transformation $f_{\mathrm{S2N}} : \mathbb{R}^{m \times d} \rightarrow \mathbb{R}^{m \times d}$ to map $z_{\mathrm{CLIP}}$ into the fMRI latent space. The ground-truth latent representation $z = \mathcal{E}(y_{\mathrm{fMRI}})$ is obtained by encoding the measured fMRI signal $y_{\mathrm{fMRI}}$ using the pretrained BrainVAE encoder $\mathcal{E}$. Finally, the model is trained by minimizing the voxel-wise mean squared error (MSE) between the aligned latent representation $z_{Align} = f_{\mathrm{S2N}}(z_{\mathrm{CLIP}})$ and the target fMRI embedding $z$:

$$\mathcal{L}_{\mathrm{S2N}} = \mathrm{MSE}(f_{\mathrm{S2N}}(z_{\mathrm{CLIP}}), z). \tag{5}$$

Compared to diffusion-based alignment, like Diffusion Prior [44] used in MindEye [47] and Diffusion Transformer [41] used in MindSimulator [4], our one-step mapping strategy eliminates the need for iterative denoising and avoids the **training–inference distribution gap** (see Appendix D) by operating entirely within a semantically structured latent space. Instead of starting from random noise, the S2N Mapper receives well-formed CLIP embeddings as input and directly predicts fMRI latent representations that lie within the target manifold. This one-step, point-to-distribution mapping eliminates the need for handcrafted priors, repeated sampling, or post-hoc averaging used in Mind-Simulator [4], facilitating a stable and computationally efficient visual-to-fMRI synthesis process (see Appendix D for details).

### 2.3 Visual-to-fMRI Inference

At inference time, our framework generates fMRI signals directly from visual stimuli through a streamlined pipeline. Given an input image $x$, we extract its semantic embedding $z_{\text{CLIP}} = \mathcal{V}(x)$ using the frozen CLIP encoder. The S2N Mapper then maps $z_{\text{CLIP}}$ to the aligned latent representation $z_{Align} = f_{\text{S2N}}(z_{\text{CLIP}})$, which is subsequently decoded by the pretrained BrainVAE decoder $\mathcal{D}(\cdot)$ to synthesize the corresponding fMRI response $\hat{y}_{\text{fMRI}} = \mathcal{D}(\hat{z})$. This inference process requires no iterative refinement, handcrafted priors, or post-selection procedures, allowing fast and stable visual-to-fMRI synthesis suitable for downstream analysis.

## 3 Experimental Settings

We conduct experiments on the Natural Scenes Dataset (NSD) [2], a large-scale fMRI dataset in which 8 subjects viewed natural images from the COCO dataset [29] across approximately 40 hours of scanning. Following MindSimulator [4], we focus on 4 subjects (Sub-1, Sub-2, Sub-5, Sub-7) who completed all experimental sessions. For each subject, we use 9,000 unique images for training and evaluate on a shared set of 1,000 test images, each presented across 3 trials to account for response variability. See Appendix A for additional dataset details.

**Implementation Details.** We adopt the pretrained OpenCLIP ViT-bigG/14 model [43] as a frozen visual encoder to extract semantic embeddings from visual stimuli. Our model is implemented and trained on 4 NVIDIA A100 GPUs (40GB memory per GPU), with training completed within 2 hours. Note that a single NVIDIA A100 GPU is sufficient to train a variant of SynBrain ($d$=1024) with trivial performance degradation (see Appendix Table 4 for details).

**Training Settings.** We explore visual-to-fMRI synthesis across two settings and assess whether the generated fMRI signals can effectively support decoding in few-shot scenarios.

**i) Subject-specific full-data (40-hour) training:** In this setting, we train BrainVAE using the AdamW optimizer with $(\beta_1, \beta_2) = (0.9, 0.999)$, a learning rate of $1 \times 10^{-4}$, weight decay of $0.05$, and a batch size of $24$. We apply *early stopping* to prevent overfitting. The S2N Mapper is optimized with identical hyperparameters and trained for 50,000 steps.

**ii) Novel-subject few-shot (1-hour) adaptation:** In this setting, we retain the same training protocol as for source subjects but adopt a parameter-efficient strategy. Specifically, we finetune the entire BrainVAE while updating only the MLP submodules in S2N Mapper's Transformer architecture.

**iii) Data augmentation for few-shot fMRI-to-Image decoding:** In this setting, we follow the few-shot experimental protocol of MindEye2, with one key modification: we augment the limited real fMRI data using synthetic fMRI signals generated from unseen images, thereby expanding the training set. This setting helps to validate the quality of synthesized fMRI signals.

**Evaluation Metrics.** We assess model performance across three complementary levels:

**i) Voxel-Level Metrics:** To quantify reconstruction fidelity, we compute three voxel-wise similarity metrics between synthesized fMRI signals and all original trials, and report the average scores across trials: mean squared error (MSE), Pearson correlation (Pearson), and cosine similarity (Cosine).

**ii) Semantic-Level Metrics:** To assess semantic quality, we use MindEye2 [49], a pretrained diffusion-based fMRI-to-image decoder. We evaluate the semantic alignment between decoded and original images using four established metrics: Inception Score (Incep) [51], CLIP similarity (CLIP) [43], EfficientNet distance (Eff) [53], and SwAV distance (SwAV) [7].

**iii) Image Retrieval Accuracy:** We measure how well fMRI signals retain semantic information by computing cosine similarity between fMRI embeddings and CLIP visual embeddings. We compare two signal sources: (i) raw fMRI (Raw) and (ii) synthetic fMRI signals (Syn) from SynBrain. Retrieval results are averaged over 30 sampled subsets of 300 candidate images.

Please refer to Appendix B for detailed descriptions of these metrics and other fMRI-to-image decoding evaluation metrics.

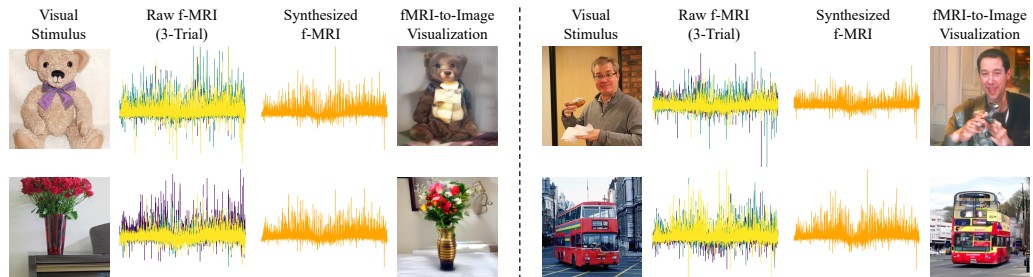

Figure 4: Visual-to-fMRI synthesis results of SynBrain and fMRI-to-image visualizations.

| Method | Voxel-Level | | | Semantic-Level (via decoding) | | | | Image Retrieval | |
|---|---|---|---|---|---|---|---|---|---|
| | MSE ↓ | Pearson ↑ | Cosine ↑ | Incep ↑ | CLIP ↑ | Eff ↓ | SwAV ↓ | Raw ↑ | Syn ↑ |
| MindSimulator (Trials=1) | .403 | .346 | - | 92.1% | 90.4% | .701 | .396 | - | - |
| MindSimulator (Trials=5) | .385 | .357 | - | 93.1% | 91.2% | .689 | .391 | - | - |
| SynBrain (Trials=1) | **.139** | **.687** | **.739** | **95.7%** | **94.3%** | **.639** | **.362** | **84.8%** | **92.5%** |
| SynBrain (Sub1→Sub2, 1h) | .160 | .619 | .675 | 89.2% | 88.1% | .751 | .431 | 19.5% | 67.4% |
| SynBrain (Sub1→Sub5, 1h) | .224 | .704 | .765 | 89.2% | 88.0% | .749 | .432 | 16.8% | 54.8% |
| SynBrain (Sub1→Sub7, 1h) | .151 | .630 | .679 | 86.8% | 84.7% | .783 | .453 | 13.2% | 76.5% |

Table 1: Quantitative visual-to-fMRI synthesis performance comparisons. Top section: Subject-specific performance averaged across 4 subjects, *Trials=N* denotes sampling repetitions during inference. Bottom section: Few-shot adaptation performance with only 1 hour of data from the novel subject (Sub2, Sub5, Sub7).

## 4 Results and Analysis

### 4.1 Main Results

**Subject-specific Visual-to-fMRI Synthesis.** We first evaluate the quality of fMRI synthesis under the subject-specific setting, where both training and evaluation are performed on the same individual. As shown in the top section of Table 1, SynBrain (Trials=1) significantly outperforms MindSimulator [4] across all voxel-level and semantic-level metrics, despite using only one-shot generation. These results highlight the effectiveness of SynBrain in producing faithful and semantically consistent neural signals without the need for repeated sampling or handcrafted priors used in MindSimulator.

Moreover, SynBrain yields higher image retrieval accuracy from synthesized fMRI signals (92.5%) than from the raw fMRI signals (84.8%). This result suggests that, while raw fMRI are inherently sparse and contain substantial redundancy, SynBrain can distill the task-relevant, semantically aligned components and **generate signals that more directly reflect high-level visual content**.

Figure 4 illustrates that even repeated trials of the same visual stimulus elicit fMRI responses with noticeable variability in fine details. SynBrain effectively learns to abstract away such fine-grained fluctuations while capturing consistent semantic patterns across trials (see Appendix E for diverse yet semantically consistent synthesis for identical visual stimulus). This is further evidenced by the fMRI-to-image visualizations via pretrained MindEye2, which reflect high-level concepts aligned with the original visual stimuli (e.g., teddy bear, man, flower, bus) despite fine-grained differences between the synthesized and raw fMRI signals.

**Few-shot Subject-adaptive Visual-to-fMRI Synthesis.** In this setting, we conduct few-shot adaptation experiments using only 1 hour of data from novel subjects (Sub2, Sub5, Sub7). As shown in the bottom section of Table 1, SynBrain demonstrates robust performance even in this low-resource setting. Notably, the adapted models maintain competitive voxel-level accuracy and high semantic consistency (e.g., CLIP: 84.7%–88.1%), despite limited supervision. Interestingly, retrieval from synthetic fMRI signals remains effective (Syn: up to 76.5%), even though raw fMRI–based retrieval performance drops significantly (Raw: 13.2%–19.5%). This semantic gap highlights the value of SynBrain-generated signals in supporting robust vision-aligned representations across individuals with minimal adaptation cost.

| Method | Low-Level | | | | High-Level | | | | Retrieval | |
|---|---|---|---|---|---|---|---|---|---|---|
| | PixCorr ↑ | SSIM ↑ | Alex-2 ↑ | Alex-5 ↑ | Incep ↑ | CLIP ↑ | Eff ↓ | SwAV ↓ | Image ↑ | Brain ↑ |
| MindEye2 (1h) | .235 | **.428** | 88.0% | 93.3% | 83.6% | 80.8% | .798 | .459 | **94.0**% | 77.6% |
| MindAligner (1h) | .226 | .415 | 88.2% | 93.3% | 83.5% | 81.8% | .800 | .459 | 90.9% | **86.9**% |
| MindEye2(1h)+DA(1h) | .243 | .419 | **90.1**% | **95.1**% | 85.1% | 84.7% | **.770** | **.432** | 87.9% | 82.0% |
| MindEye2(1h)+DA(2h) | .244 | .418 | 89.9% | **95.1**% | 85.6% | **85.0**% | .773 | .434 | 84.1% | 80.5% |
| MindEye2(1h)+DA(4h) | **.246** | .422 | **90.1**% | 94.6% | **85.8**% | 84.0% | .772 | .433 | 79.4% | 77.7% |

Table 2: Comparison of fMRI-to-image decoding performance under few-shot adaptation on Subject 1. MindEye2 and MindAligner are finetuned using only 1 hour of real fMRI data from Subject 1, while our data augmentation (DA) strategy enhances the training set with synthetic fMRI signals generated by SynBrain (Sub2→Sub1) from novel images in unseen sessions.

| Method | Voxel-Level | | | Semantic-Level (via decoding) | | | | Image Retrieval | |
|---|---|---|---|---|---|---|---|---|---|
| | MSE ↓ | Pearson ↑ | Cosine ↑ | Incep ↑ | CLIP ↑ | Eff ↓ | SwAV ↓ | Raw ↑ | Syn ↑ |
| SynBrain | **.079** | **.715** | **.769** | **96.7**% | **95.9**% | **.619** | **.350** | **94.7**% | **99.3**% |
| w/o Variation Sampling | .086 | .687 | .712 | 88.5% | 86.7% | .688 | .398 | 90.2% | 88.4% |
| w/o Contrastive Learning | .127 | .635 | .673 | 86.2% | 84.5% | .694 | .426 | 0.5% | 0.4% |
| w/o S2N Mapper | .105 | .564 | .661 | 77.3% | 75.0% | .838 | .514 | 94.7% | 50.5% |

Table 3: Ablation experiments of SynBrain under the subject-specific setting on Subject 1.

**Data Augmentation for Few-shot fMRI-to-Image Decoding** To improve fMRI-to-image decoding performance in data-limited conditions, we introduce a Data Augmentation (DA) strategy that *supplements real fMRI recordings with synthetic fMRI signals generated by SynBrain*. Specifically, given only 1 hour of real fMRI data from an unseen subject, we synthesize additional fMRI data using novel images from unseen sessions and add them to the training set. We evaluate MindEye2 (1h) augmented with varying amounts of synthetic data (1h, 2h, and 4h), and compare it against two baselines: MindEye2 (1h) and MindAligner (1h), finetuned exclusively on 1 hour of real fMRI data from Subject 1. Here, the Sub2→Sub1 model of SynBrain is used to synthesize fMRI samples.

Table 2 reports quantitative results across multiple fMRI-to-image reconstruction and retrieval metrics. Despite using the same 1-hour real dataset, models augmented with synthetic signals consistently outperform the baselines, except for the image retrieval performance. Notably, *MindEye2+DA(1h)* achieves a higher CLIP similarity (84.7% vs. 80.8%), better Inception Score (85.1 vs. 83.6), and stronger brain retrieval performance (82.0% vs. 77.6%). Interestingly, the benefit plateaus or slightly declines with more synthetic data, suggesting that *moderate augmentation yields* the *best balance between diversity and quality*. These findings demonstrate that our high-quality synthetic fMRI signals can serve as an effective form of data augmentation, improving fMRI-to-image decoding in few-shot settings without requiring time-consuming and laborious data collection procedures.

## 4.2 Ablation Study

To assess the contribution of key components in SynBrain, we conduct a series of controlled ablation experiments on Subject 1, examining the effects of removing the variational sampling ($\mathcal{L}_{KL}$), contrastive learning ($\mathcal{L}_{CLIP}$), and S2N Mapper. Results are summarized in Table 3.

**Impact of Variation Sampling:** Removing this component (*i.e.*, using a deterministic autoencoder) significantly degrades semantic-level alignment (Incep: 96.7% → 88.5%, CLIP: 95.9% → 86.7%). This highlights the importance of distribution-level probabilistic learning for capturing functional consistency behind the inherent variability of neural responses rather than overfitting to specific deterministic patterns.

**Impact of Contrastive learning:** Removing this component causes semantic retrieval accuracy to collapse (99.3% → 0.5%), indicating that the BrainVAE encoder fails to map neural responses into a shared visual-semantic space. Interestingly, the model still retains non-trivial semantic alignment performance (e.g., CLIP: 84.5%), which suggests that BrainVAE implicitly captures structural and semantic regularities in the fMRI data during reconstruction. The S2N Mapper, trained in the second stage, can leverage this structured latent space to partially align with visual semantics, despite the absence of contrastive guidance.

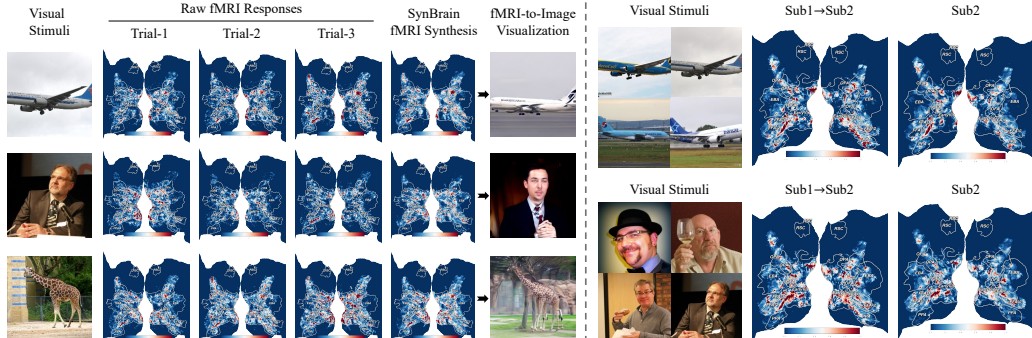

Figure 5: Cross-trial and cross-subject brain functional consistency visualization. **Left:** Comparisons of activation maps between different fMRI trials and our synthesized fMRI evoked by the same stimuli. **Right:** Comparisons of activation maps between Sub2 (Full-data, 40h) and Sub1→Sub2 (Few-Shot, 1h) evoked by representative categories of visual stimuli.

**Impact of S2N Mapper:** The S2N Mapper is introduced after contrastive alignment to project CLIP embeddings directly into the fMRI latent manifold. Eliminating this module leads to the most pronounced semantic degradation (CLIP: 95.9% → 75.0%; Inception: 96.7% → 77.3%), highlighting its essential role in bridging the modality gap.

### 4.3 Brain Functional Analysis

Neural responses exhibit inherent variability while maintaining consistent functions—a fundamental property of biological systems that our probabilistic approach explicitly models [14]. Analyzing SynBrain's ability to capture this duality offers deeper insight into neural information processing.

**Cross-trial Functional Consistency.** Neural responses to identical stimuli vary across trials due to physiological noise, spontaneous fluctuations, and intrinsic stochastic properties [36, 9, 42]. Our analysis of synthesized versus raw fMRI responses (Figure 5) reveals an important principle: category-selective regions (e.g., fusiform face area) maintain consistent activation patterns across trials despite substantial voxel-level variability [14, 40]. SynBrain successfully captures this balanced relationship between variability and consistency, suggesting that trial-to-trial fluctuations follow a structured pattern rather than reflecting random noise [12]. The preservation of regional activation amid voxel-level fluctuations supports a hierarchical organization of neural variability—while low-level sensory details may vary, high-level semantic representations remain stable. This aligns with theoretical frameworks proposing that neural variability serves as a form of probabilistic computation [40], enabling the brain to represent multiple interpretations of input while converging on consistent semantics.

**Cross-subject Functional Consistency.** Despite individual differences in brain anatomy and activity, similar representational dynamics emerge when processing similar stimuli [45, 65]. Our cross-subject adaptation results provide a computational explanation for this phenomenon. When adapting from Subject-1 to Subject-2 using only 1 hour of data, SynBrain produces activation patterns closely aligned with those obtained from a model trained on the full Subject-2 dataset, particularly in category-selective regions. This efficient adaptation suggests that individual variability in visual processing is highly structured, likely constrained to a low-dimensional subspace that minimally interferes with core semantic representations. The success of our adaptation approach supports the hypothesis that invariant semantic representations can be effectively disentangled from subject-specific characteristics [14, 40, 65]. This principle may explain recent findings of preserved neural dynamics across individuals and species [45], suggesting a fundamental organizational mechanism by which the brain balances individual variation with functional consistency.

## 5 Related Work

**Visual-to-fMRI Encoding.** Encoding visual input into fMRI responses has been a major topic of interest in neuroscience and computational modeling [37, 24, 18, 54]. Conventional methods typically adopt regression-based frameworks to establish mappings from visual features to voxel-

wise neural activity. Some studies aim to improve the quality of input features through task-specific models [20, 56], or by optimizing the selection of visual stimuli [31], while others focus on enhancing the regression function itself through nonlinear architectures [16, 61, 1, 27, 33, 5] However, most of these models predict a single deterministic response for each image, which limits their ability to capture the variability inherent in neural data.

More recently, MindSimulator [4] proposed a generative encoding approach using a diffusion-based model built on a deterministic MLP-based autoencoder. However, its stochasticity is introduced solely at inference time via diffusion sampling started from a Gaussian noise distribution, while the generative process still remains deterministic. As such, the system does not learn a latent distribution over neural responses, but instead treats random perturbations as input variations, requiring averaged multiple samplings and heuristic post-selection to produce stable and plausible results.

The proposed *SynBrain* framework addresses these limitations by employing probabilistic learning to model neural variability and introducing a one-step point-to-distribution mapping mechanism, allowing stable and semantically aligned fMRI synthesis in a single forward pass.

**Few-Shot fMRI-to-Visual Decoding.** The task of fMRI-to-visual decoding aims to reconstruct visual stimuli from recorded fMRI responses, with related neural decoding efforts also explored in the Electroencephalography (EEG) [19, 30, 64, 59] and Magnetoencephalography (MEG) [6, 62] domains. Recent approaches typically map fMRI signals into pretrained vision-language embedding spaces (e.g., CLIP) and decode them into images using powerful generative backbones [8, 35, 52, 47]. While subject-specific decoding has achieved impressive results, however, collecting high-quality fMRI data is expensive, time-consuming, and subject to strong individual variability—making large-scale subject-specific training impractical in many scenarios. To address this, recent studies have explored *cross-subject few-shot adaptation*, where models are adapted to novel individuals using only 1 hour of brain recordings [48, 46, 57, 17].

In this setting, MindEye2 [46] employs ridge regression to align subjects into a shared latent space, followed by a universal decoding module. MindAligner [11] further introduces an explicit brain functional alignment model that relaxes the constraint of shared stimuli, achieving state-of-the-art decoding performance with minimal learned parameters.

Despite their promise, these methods remain challenged by inter-subject variability under limited supervision. In this work, we investigate *whether synthetic fMRI signals can serve as an effective form of data augmentation*—expanding the number of fMRI–image training pairs and thereby improving visual decoding performance in few-shot cross-subject adaptation settings.

## 6   Conclusions

SynBrain advances visual encoding models by explicitly addressing the biological reality of neural variability while preserving functional consistency. Our latent space modeling approach, integrating BrainVAE for cross-trial variability modeling and S2N for stimulus-to-neural distribution mapping, enables biologically plausible neural response modeling. Experimental results demonstrate superior encoding performance while maintaining neural response characteristics, establishing a more accurate digital twin of visual neural circuits.

**Broader Impact.** SynBrain's probabilistic modeling paradigm extends naturally to other neural modalities and brain regions beyond the visual cortex. This approach has particular potential for brain-computer interface applications, where modeling individual variability while maintaining functional consistency could significantly reduce calibration requirements and improve robustness.

**Limitations and Future Direction.** Despite these advances, several challenges remain. First, our reliance on pre-trained vision models may introduce inherited representational biases that may not perfectly align with neural processing. Second, while we model response variability, we cannot yet account for all variance sources, such as attentional state fluctuations or neuromodulatory effects. These limitations highlight opportunities for further refinement of probabilistic neural encoding approaches. Our findings reveal that individual neural differences occupy structured low-dimensional subspaces largely orthogonal to semantic representations, explaining SynBrain's efficient cross-subject transfer capability. This computational principle enables personalized neurotechnology with minimal calibration data. Future directions include extending this framework to other sensory modalities and integrating temporal dynamics for multimodal brain modeling, with potential applications spanning fundamental neuroscience research and clinical brain-computer interfaces.

## Acknowledgments

This work was supported by Shanghai Artificial Intelligence Laboratory and the JC STEM Lab of AI for Science and Engineering, funded by The Hong Kong Jockey Club Charities Trust, the Research Grants Council of Hong Kong (Project No. CUHK14213224).

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

# A    NSD Dataset

In this study, we leverage the largest publicly available fMRI-image dataset, the Natural Scenes Dataset (NSD), which encompasses extensive 7T fMRI data collected from eight subjects while they viewed images from the COCO dataset. Each subject viewed each image for 3 seconds and indicated whether they had previously seen the image during the experiment. Our analysis focuses on data from four subjects (Sub-1, Sub-2, Sub-5, and Sub-7) who completed all viewing trials. The training dataset consists of 9,000 images and 27,000 fMRI trials, while the test dataset includes 1,000 images and 3,000 fMRI trials, with up to 3 repetitions per image. It is important to note that the test images are consistent across all subjects, whereas distinct training images are utilized.

We used preprocessed scans from NSD for functional data, with a resolution of 1.8 mm. Our analysis involved employing single-trial beta weights derived from generalized linear models, along with region-of-interest (ROI) data specific to early and higher (ventral) visual regions as provided by NSD. The ROI voxel counts for the respective four subjects are as follows: [15724, 14278, 13039, 12682]. Detailed fMRI preprocessing procedures and additional information can be found in the source paper [2] and on the NSD website[3].

# B    Evaluation Metric

## B.1    fMRI-to-Image Decoding Model.

**i) Subject-specific setting:**  All semantic-level metrics are obtained by decoding the synthesized fMRI using a fixed *MindEye2* model trained on the corresponding subject using **_40 hours_** of data.

**ii) Few-shot adaptation setting:**  All semantic-level metrics are obtained by decoding the synthesized fMRI using a fixed *MindEye2* model trained on the corresponding subject using **_1 hour_** of data.

## B.2    Visual-to-fMRI Synthesis Metrics

We provide detailed descriptions of the evaluation metrics used to assess SynBrain's performance in visual-to-fMRI synthesis across voxel-level accuracy, semantic-level alignment, and cross-modal retrieval capability.

**Voxel-Level Metrics.**    To evaluate the fidelity of synthesized fMRI signals, we compare the generated output with ground-truth fMRI recordings for each stimulus on a voxel-by-voxel basis. The following three metrics are computed and averaged across all test fMRI trials: i) **MSE**: Computes the mean squared error between synthetic and ground-truth fMRI signals; ii) **Pearson:** Measures the linear Pearson correlation coefficient between synthetic and ground-truth fMRI signals; iii) **Cosine:** Evaluates the angle-based cosine similarity between synthetic and ground-truth fMRI signals.

**Semantic-Level Metrics via fMRI-to-Image Decoding.**    To assess whether the synthesized fMRI signals preserve the underlying semantic content of the input stimulus, we leverage the pretrained fMRI-to-image decoder MindEye2 [49] to transform synthetic fMRI into images. The decoded images are then compared with the original ground-truth images using multiple semantic metrics: i) **Incep:** A two-way comparison of the last pooling layer of InceptionV3; ii) **CLIP:** A two-way comparison of the output layer of the CLIP-Image model; iii) **Eff:** A distance metric gathered from EfficientNet-B1 model; iv) **SwAV:** A distance metric gathered from SwAV-ResNet50 model.

A two-way comparison evaluates the accuracy percentage by determining whether the original image embedding aligns more closely with its corresponding brain embedding or with a randomly selected brain embedding.

**Image Retrieval Accuracy.**    We evaluate cross-modal retrieval performance between CLIP visual embeddings and fMRI embeddings from two sources: i) **Raw:** The raw fMRI signal from the test set; ii) **Syn:** The synthetic fMRI signal of SynBrain from the corresponding visual input. We compute the cosine similarity between fMRI embeddings encoded by BrainVAE and 300 candidate image embeddings extracted from test images, with one being the ground-truth visual stimulus for the fMRI

---

[3]https://naturalscenesdataset.org

---

**Algorithm 1** BrainVAE Architecture

---

1: **Input:** fMRI signal $x$, CLIP visual representation $z_{\text{CLIP}}$
2: **Encoder:**
3:     $h = \texttt{Encoder}(x)$                                                           $\triangleright$ Hierarchical Conv+Attn backbone
4: **Pre-projector (MLP) for Mean or LogVar:**
5:     $h = \texttt{LayerNorm}(h) \rightarrow \texttt{GELU}$
6:     $h = \texttt{Linear}(4096 \rightarrow d) \rightarrow \texttt{LayerNorm} \rightarrow \texttt{GELU}$                               $\triangleright d = 2048$
7:     $h = \texttt{Linear}(d \rightarrow d) \rightarrow \texttt{LayerNorm} \rightarrow \texttt{GELU}$
8:     $\mu$ or $\log \sigma^2 = \texttt{Linear}(d \rightarrow 1664)$
9: **Sampling:**
10:     $z \sim \mathcal{N}(\mu, \ \sigma^2)$
11: **Post-Projector (MLP):**
12:     $z = \texttt{LayerNorm}(z) \rightarrow \texttt{GELU}$
13:     $z = \texttt{Linear}(1664 \rightarrow d) \rightarrow \texttt{LayerNorm} \rightarrow \texttt{GELU}$                            $\triangleright d = 2048$
14:     $z = \texttt{Linear}(d \rightarrow d) \rightarrow \texttt{LayerNorm} \rightarrow \texttt{GELU}$
15:     $h' = \texttt{Linear}(d \rightarrow 4096)$
16: **Decoder:**
17:     $\hat{x} = \texttt{Decoder}(h', V_s)$                 $\triangleright V_s$ denotes the voxel count for subject $S$
18: **Loss:**
19:     $\mathcal{L} = \texttt{MSELoss}(x, \hat{x}) + \texttt{KLWeight} \cdot \texttt{KL}(\mu, \sigma^2) + \texttt{CLIPWeight} \cdot \texttt{CLIPLoss}(z, z_{\text{CLIP}})$
20: **return** $\mathcal{L}$

---

data. Retrieval performance is evaluated by calculating the average Top-1 retrieval accuracy (with a chance level of 1/300) and repeating the process 30 times to account for batch sampling variability.

### B.3 Few-shot fMRI-to-Image Decoding Metrics

We follow MindEye2 [49] to evaluate few-shot fMRI-to-image decoding performance in low-level, high-level, and brain-image cross-modal retrieval.

**Low-Level Metrics.** These metrics assess fundamental perceptual image features, providing insights into the visual content and structure of the image, measured by: i) **PixCorr** [10]: Pixel-level correlation between reconstructed and test images; ii) **SSIM** [58]: Structural similarity index; iii) **AlexNet** [25]: Alex-2 and Alex-5 are the 2-way comparisons of the second (early) and fifth (middle) layers of AlexNet, respectively.

**High-Level Metrics.** Same as *Semantic-Level Metrics via fMRI-to-Image Decoding* in Section B.2.

**Retrieval Metrics.** For **image retrieval**, we compute the cosine similarity between the predicted fMRI embedding and 300 candidate image embeddings extracted from test images, with one being the ground-truth visual stimulus for the fMRI data. Retrieval performance is evaluated by calculating the average Top-1 retrieval accuracy (with a chance level of 1/300) and repeating the process 30 times to account for batch sampling variability. For **brain retrieval**, we follow the same procedure, but with brain and image samples interchanged.

## C  Network Architecture

### C.1  BrainVAE Architecture

We provide detailed specifications of the BrainVAE architecture, including the encoder, projector modules, and decoder.

**Overall Architecture.** As shown in Algorithm 1, BrainVAE consists of an encoder, two projection modules (for mean and log-variance), a stochastic sampling step, a post-projector, and a decoder. The input fMRI signal $x \in \mathbb{R}^{1 \times V_s}$ is first encoded into a hidden feature $h \in \mathbb{R}^{256 \times 4096}$ through a multi-resolution convolutional encoder. The mean $\mu$ and log-variance $\log \sigma^2$ of the latent Gaussian distribution are obtained by passing $h$ through two identical MLP-based pre-projectors. Each projector

---

**Algorithm 2** BrainVAE Encoder Architecture

---

1: **Input:** fMRI signal $x \in \mathbb{R}^{1 \times V_s}$          ▷ $V_s$ denotes the voxel count for subject $S$
2: $x \leftarrow \texttt{Conv1D}(x, \texttt{out\_channels} = 128, \texttt{kernel} = 1, \texttt{stride} = 1, \texttt{padding} = 0)$
3: $x \leftarrow \texttt{AdaptiveMaxPool1D}(x, \texttt{output} = 8192)$
4: $H \leftarrow [x]$          ▷ initialize hidden state list
5: **for** $i = 1$ to $num\_blocks$ **do**          ▷ num_blocks=3, ch_mult=[1, 2, 4, 4]
6:      $c_{\text{in}} = 128 \times \texttt{ch\_mult}[i - 1]$
7:      $c_{\text{out}} = 128 \times \texttt{ch\_mult}[i]$
8:      **for** $j = 1$ to $num\_res\_blocks$ **do**          ▷ num_res_blocks = 2
9:          $h \leftarrow \texttt{ResnetBlock}(H[-1], c_{\text{in}}, c_{\text{out}})$
10:          Append $h$ to $H$
11:      **end for**
12:      **if** $i \leq num\_down\_blocks$ **then**          ▷ num_down_blocks = 1
13:          $h \leftarrow \texttt{Downsample}(H[-1])$
14:          Append $h$ to $H$
15:      **end if**
16: **end for**
17: $h \leftarrow H[-1]$
18: $h \leftarrow \texttt{ResnetBlock}(h) \rightarrow \texttt{SelfAttention}(h) \rightarrow \texttt{ResnetBlock}(h)$          ▷ Middle block
19: $h \leftarrow \texttt{LayerNorm}(h) \rightarrow \texttt{SiLU}(h)$
20: $h \leftarrow \texttt{Conv1D}(h, \texttt{out\_channels} = 256, \texttt{kernel} = 1, \texttt{stride} = 1, \texttt{padding} = 0)$
21: **return** $h \in \mathbb{R}^{256 \times 4096}$

---

begins with LayerNorm and GELU activation, followed by two hidden Linear layers with configurable dimension $d$ (default $d = 2048$), and ends with a final projection to $\mathbb{R}^{256 \times 1664}$.

A latent representation $z \sim \mathcal{N}(\mu, \sigma^2)$ is then sampled and passed through a post-projector of identical structure, which maps $z \in \mathbb{R}^{256 \times 1664}$ back to $h' \in \mathbb{R}^{256 \times 4096}$. The decoder takes $h'$ and reconstructs the fMRI signal $\hat{x} \in \mathbb{R}^{1 \times V_s}$, where $V_s$ denotes the subject-specific voxel count for subject $S$.

The training objective combines three components: (1) an $\ell_2$ reconstruction loss between $x$ and $\hat{x}$, (2) KL divergence between the posterior and prior distributions, and (3) a contrastive loss aligning the latent $z$ with the CLIP embedding $z_{\text{CLIP}}$ of the visual stimulus.

**Encoder Architecture.** The encoder design is detailed in Algorithm 2. The input fMRI signal $x \in \mathbb{R}^{1 \times V_s}$ first passes through a 1D convolution layer with 128 output channels and a kernel size of 1, followed by adaptive max pooling to a fixed length of 8192. This allows the model to process inputs from different subjects without introducing subject-specific parameters.

The core of the encoder is a hierarchical ResNet-style backbone with three resolution levels, specified by $\texttt{ch\_mult} = [1, 2, 4, 4]$. Each level includes two residual blocks, and downsampling is applied only after the first level (i.e., $\texttt{num\_down\_blocks} = 1$). A middle block with a self-attention layer is inserted between two additional residual blocks. After the final ResBlock, the feature is normalized, activated using SiLU, and projected via a 1D convolution to produce the output $h \in \mathbb{R}^{256 \times 4096}$.

**Decoder Architecture.** The decoder mirrors the encoder in a symmetric fashion. It takes the post-projected feature $h' \in \mathbb{R}^{256 \times 4096}$ as input and processes it through a hierarchical ResNet-style upsampling backbone. The initial convolutional layer increases the feature dimensionality to match the resolution of the encoder output. Then, residual blocks and upsampling layers are applied across three resolution levels (reversing the encoder's channel multipliers $\texttt{ch\_mult} = [4, 4, 2, 1]$). One upsampling operation is performed at the final resolution stage (i.e., $\texttt{num\_up\_blocks} = 1$), consistent with the encoder's downsampling configuration.

A middle block, identical in structure to the encoder's, includes a self-attention layer and two ResNet blocks. After the final upsampling stage, the decoder produces an intermediate sequence of length 8192. Since the number of voxels $V_s$ may differ across subjects, we apply a 1D linear interpolation step to resample the sequence to exactly match the target length $V_s$. A final 1D convolution with output channels = 1 maps the feature to the reconstructed fMRI signal $\hat{x} \in \mathbb{R}^{1 \times V_s}$.

**Ablation Study on Latent Dimensionality.** To investigate the effect of latent dimensionality on synthesis performance, we conduct an ablation study by varying the number of downsampling blocks in BrainVAE from 1 to 3. This progressively reduces the latent dimension of the fMRI representation from 4096 to 2048 and 1024, respectively. As shown in Table 4, using only one down block (i.e., d=4096) consistently achieves the best performance across voxel-level, semantic-level, and retrieval metrics. In contrast, deeper compression leads to a gradual decline in synthesis quality. These results suggest that preserving higher-dimensional latent features helps retain fine-grained temporal structure in fMRI signals, which is critical for semantic fidelity and voxel-level reproduction.

| Blocks | Voxel-Level | | | Semantic-Level (via decoding) | | | | Retrieval | |
|---|---|---|---|---|---|---|---|---|---|
| | MSE ↓ | Pearson ↑ | Cosine ↑ | Incep ↑ | CLIP ↑ | Eff ↓ | SwAV ↓ | Raw ↑ | Syn ↑ |
| 1 (d=4096)* | **.079** | **.715** | **.769** | **96.7%** | **95.9%** | **.619** | **.350** | **94.7%** | **99.3%** |
| 2 (d=2048) | .082 | .703 | .746 | 95.6% | 95.0% | .622 | .356 | 94.1% | 98.9% |
| 3 (d=1024) | .087 | .701 | .733 | 95.2% | 94.5% | .643 | .363 | 93.5% | 98.0% |

Table 4: Quantitative visual-to-fMRI synthesis performance with different numbers of down blocks in BrainVAE under subject-specific setting on Subject 1. * means the final setting adopted in this work.

## C.2 S2N Mapper Architecture

The **S2N (Stimulus-to-Neural) Mapper** is implemented as a Transformer encoder that maps CLIP visual embeddings to fMRI latent representations. We use an 8-layer Transformer with 13 attention heads per layer. The input to the mapper is the CLIP visual representation $z_{\text{CLIP}} \in \mathbb{R}^{256 \times 1664}$, where 256 is the number of tokens and 1664 is the token embedding dimension derived from the OpenCLIP bigG model. To retain positional information, we use fixed sinusoidal positional encodings, which are added to the input before the first self-attention layer. Each Transformer block follows a pre-layer normalization design and includes multi-head self-attention, a feed-forward MLP with GELU activation, and residual connections.

The S2N Mapper performs a point-to-distribution transformation by mapping visual embeddings to the center of fMRI latent distributions $z_{\text{fMRI}} \in \mathbb{R}^{256 \times 1664}$, which can be used directly or after sampling to synthesize realistic and semantically aligned fMRI responses.

**Ablation Study on Number of Attention Layers.** We investigate how the number of Transformer layers in the S2N Mapper affects visual-to-fMRI synthesis quality. As shown in Table 5, we compare 4, 8, and 12-layer configurations under the subject-specific setting on Subject 1. The 8-layer model achieves the best overall performance, while increasing the number of layers to 12 does not yield further improvements. Hence, we adopt the 8-layer configuration as the final setting in this work.

| Layer | Voxel-Level | | | Semantic-Level (via decoding) | | | | Retrieval | |
|---|---|---|---|---|---|---|---|---|---|
| | MSE ↓ | Pearson ↑ | Cosine ↑ | Incep ↑ | CLIP ↑ | Eff ↓ | SwAV ↓ | Raw ↑ | Syn ↑ |
| 4 | .086 | .704 | .750 | 95.8% | 95.2% | .632 | .358 | 94.7% | 98.5% |
| 8* | **.079** | **.715** | **.769** | **96.7%** | 95.9% | **.619** | **.350** | 94.7% | **99.3%** |
| 12 | .080 | **.715** | .768 | 96.6% | **96.0%** | .620 | **.350** | 94.7% | **99.3%** |

Table 5: Quantitative visual-to-fMRI synthesis performance with different numbers of attention layers in S2N Mapper under the subject-specific setting on Subject 1. * indicates the final setting adopted in this work.

## D  Comparison between Diffusion Transformer and S2N Mapper

**Distribution Mismatch in Diffusion Transformer.** MindSimulator [4] leverages the Diffusion Transformer (DiT) [41] to generate fMRI representations conditioned on visual prompts through iterative denoising. However, as shown in Figure 6, we observe a significant distributional mismatch between training and inference phases in this setup.

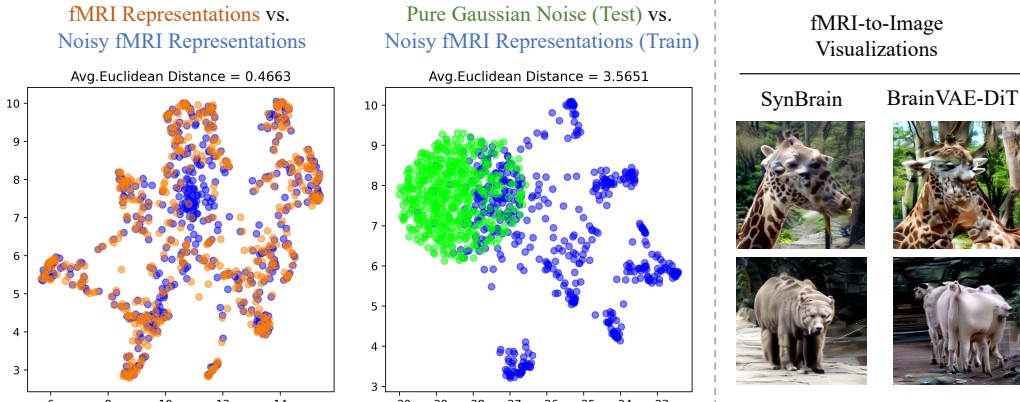

Figure 6: UMAP visualizations of *distribution gap* in DiT and fMRI-to-image visualizations. **Left:** Noisy fMRI representations (blue) used for DiT training still lie close to the original fMRI representations (orange), but far away from pure Gaussian noise (green) used for DiT inference, showing a clear distribution gap between training and testing stages in DiT. **Right:** SynBrain (BrainVAE-S2N, one-step mapping started from original fMRI representations) produces more realistic and semantically consistent images compared to BrainVAE-DiT (multi-step denoising started from Gaussian noise).

| Method | Voxel-Level | | | Semantic-Level (via decoding) | | | | Retrieval | |
|---|---|---|---|---|---|---|---|---|---|
| | MSE ↓ | Pearson ↑ | Cosine ↑ | Incep ↑ | CLIP ↑ | Eff ↓ | SwAV ↓ | Raw ↑ | Syn ↑ |
| BrainVAE-DiT | .088 | .689 | .748 | 94.8% | 93.5% | .642 | .376 | 94.7% | 93.2% |
| SynBrain | **.079** | **.715** | **.769** | **96.7%** | **95.9%** | **.619** | **.350** | 94.7% | **99.3%** |

Table 6: Quantitative comparison of fMRI synthesis performance between SynBrain (BrainVAE-S2N) and BrainVAE-DiT under a subject-specific setting on Subject 1.

i) **Training:** During training, DiT learns to denoise noisy fMRI latent representations formed by adding Gaussian noise to ground-truth fMRI signals. As visualized in Figure 6-Left, these noisy representations (blue) still lie close to the original fMRI manifold (orange), maintaining structural similarity (Avg. Euclidean Distance = 0.4663).

ii) **Inference:** At test time, however, the model must denoise pure Gaussian noise (green), which is far from the fMRI manifold (blue; Figure 6-Middle). The resulting distribution gap (Avg. Euclidean Distance = 3.5651) leads to unstable generation quality and typically requires multiple samples and heuristic post-selection.

**S2N Mapper for Direct Semantic Alignment.** To address this distribution gap, we propose replacing the diffusion transformer with a one-step Semantic-to-Neural (S2N) Transformer Mapper. The resulting model, SynBrain (BrainVAE-S2N), directly maps to the fMRI latent distribution from CLIP embeddings without iterative sampling. This approach ensures distributional consistency and semantic grounding throughout the generation process.

**Performance Comparison.** Table 6 compares BrainVAE-DiT with SynBrain (BrainVAE-S2N) in the task of visual-to-fMRI synthesis. Both models share the same VAE-based latent architecture, but differ in how visual representations are mapped to fMRI latent distributions.

SynBrain consistently outperforms BrainVAE-DiT across all evaluation levels. It yields higher semantic fidelity as measured by Incep (96.7% vs. 94.8%) and CLIP similarity (95.9% vs. 93.5%), and significantly improves retrieval accuracy of synthetic fMRI signals (Syn: 99.3% vs. 93.2%). Figure 6-Right presents qualitative examples from both models. SynBrain produces more realistic and semantically coherent images than BrainVAE-DiT, which occasionally exhibits semantic drift, likely due to initialization from unstructured Gaussian noise. These results demonstrate that SynBrain more effectively preserves the semantic content of visual inputs in the synthesized fMRI representations. By

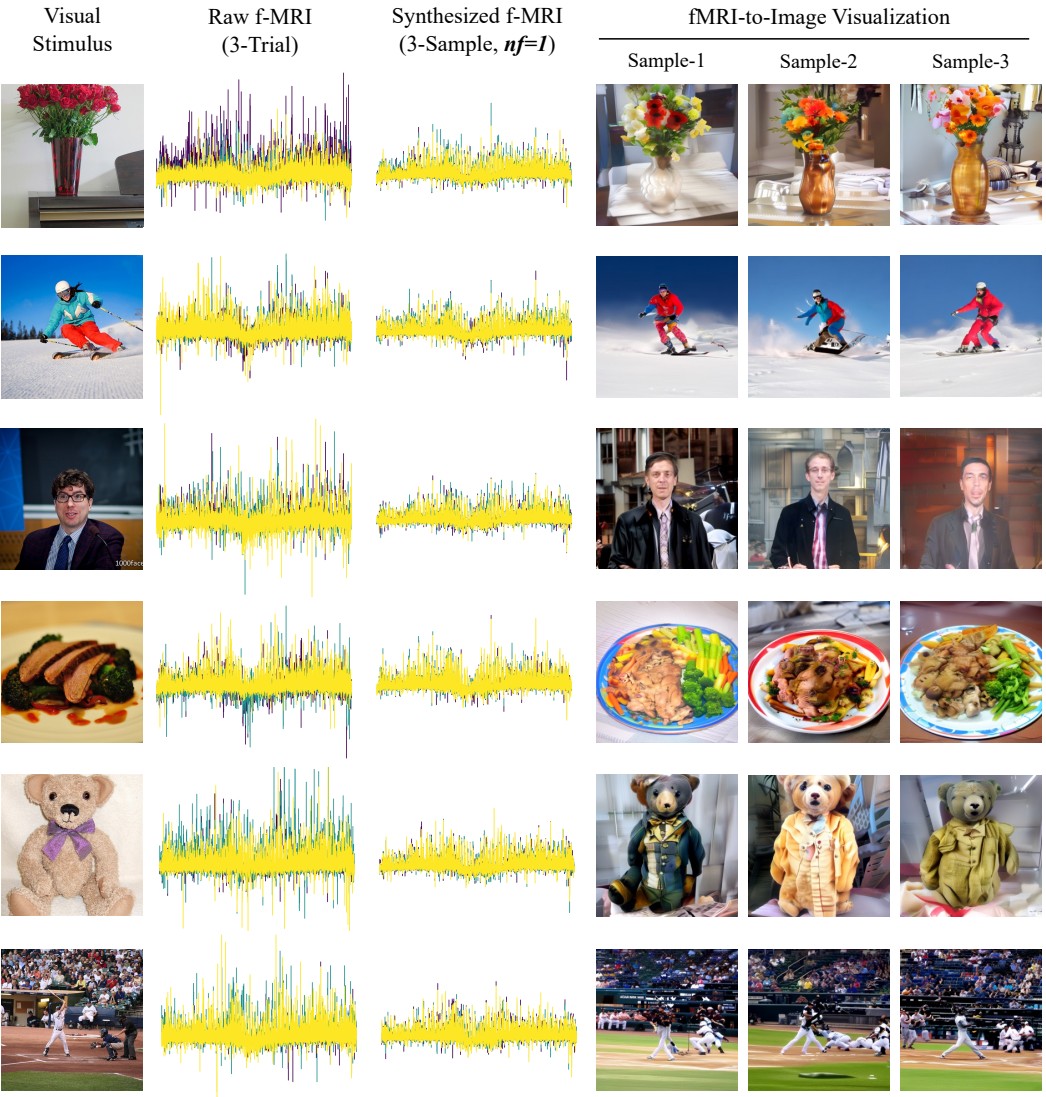

Figure 7: Semantically consistent synthesis of SynBrain under stochastic sampling with **nf=1**.

eliminating the distribution mismatch introduced by denoising from noise, the S2N mapper provides a more stable, interpretable, and semantically grounded solution for visual-to-fMRI synthesis.

## E    Semantically Consistent Synthesis under Stochastic Sampling

To evaluate the generative robustness of **SynBrain**, we investigate its ability to synthesize diverse yet semantically consistent fMRI responses by sampling from the latent distribution. Given a distributional center $z_{\text{Align}}$ predicted by the **S2N Mapper**, we introduce stochastic variation by adding Gaussian noise scaled by a noise factor $nf$, such that:

$$z = z_{\text{Align}} + nf \cdot \epsilon, \quad \epsilon \sim \mathcal{N}(0, I)$$

Figure 7 presents the synthesis results for a strong noise setting, $nf = 1$. SynBrain can produce fMRI responses that yield highly consistent visual outputs across multiple samples. This suggests that the learned latent space is semantically structured and resilient to stochastic perturbations. Empirically, reducing the noise factor ($nf < 1$) leads to more stable fMRI synthesis with reduced variability. When $nf = 0$, sampling is disabled and SynBrain directly uses the predicted distributional center $z_{\text{Align}}$, which may produce a single but more realistic and faithful response.

| Method | Visual Encoder | Voxel-Level | | | Semantic-Level (via decoding) | | | | Image Retrieval | |
|---|---|---|---|---|---|---|---|---|---|---|
| | | MSE ↓ | Pearson ↑ | Cosine ↑ | Incep ↑ | CLIP ↑ | Eff ↓ | SwAV ↓ | Raw ↑ | Syn ↑ |
| MindSimulator | CLIP | .417 | .326 | – | 92.8% | 89.8% | .714 | .402 | – | – |
| SynBrain | **CLIP\*** | **.079** | **.715** | **.769** | **96.7**% | **95.9**% | **.619** | **.350** | **94.7**% | **99.3**% |
| | MAE | .097 | .676 | .737 | 91.0% | 90.5% | .705 | .389 | 68.6% | 83.7% |
| | DINOv2 | .089 | .679 | .728 | 92.6% | 91.8% | .701 | .375 | 76.3% | 87.6% |

Table 7: Visual-to-fMRI synthesis performance with various visual encoders.

These results demonstrate that SynBrain supports structured one-to-many visual-to-fMRI generation and allows controllable sampling of plausible neural responses under varying uncertainty conditions.

## F  Vision Model Comparison

In this section, we evaluate the impact of visual encoder choice by replacing CLIP in SynBrain with two advanced visual foundation models, DINOv2 [39] and MAE [22]. As shown in the table below, CLIP achieves the best overall performance across voxel-level and semantic levels. We attribute this to its vision-language contrastive objective, which provides stronger semantic supervision compared to purely visual models like DINOv2 and MAE. Nonetheless, the relatively strong performance of these alternatives indicates that SynBrain remains robust across different encoder choices.

We also include a comparison with MindSimulator (5-Trial), which likewise uses CLIP as its visual encoder. Notably, MindSimulator averages results over five generated fMRI trials to improve performance, whereas SynBrain achieves superior results with just a single trial. This suggests that SynBrain benefits not only from a strong encoder but also from a more effective model design and training strategy.

## G  Additional Baselines

**Subject-specific Baselines.**   Here we conduct additional comparisons with two representative baselines: (i) **GNet** [50], and (ii) **Linear Regression** (LinearReg), a classical deterministic voxel-wise visual encoding model, using a simple two-layer linear architecture with a 4,096-dimensional hidden space. As shown in Table 8, SynBrain achieves consistently **superior results across multiple levels**, highlighting its effectiveness in fMRI synthesis with both voxel-level structure and semantic consistency.

**Subject-adaptive baselines.**   Taking MindSimulator [4] as a cross-subject adaptation baseline is limited by two practical constraints. i) MindSimulator is not open-sourced and incorporates several hand-crafted components (e.g., resting-state initialization, custom noise injection), making it difficult to reproduce; ii) MindSimulator's Autoencoder is subject-specific and tightly coupled to fixed voxel dimensions, rendering it incompatible with cross-subject scenarios where ROI sizes vary. These limitations prevent a direct and meaningful comparison between SynBrain and MindSimulator in the context of cross-subject adaptation.

To establish a transparent and reliable baseline, we implemented Linear Regression (LinearReg), the most classical encoding model, using a simple two-layer linear architecture. The first layer maps CLIP features to a 4,096-dimensional hidden space, and the second layer projects to subject-specific voxel responses. During few-shot adaptation, we freeze the first layer and train only the second layer using 1-hour data from the target subject. As shown in Table 1 and Table 9, SynBrain consistently outperforms Linear Regression across voxel-wise correlation, semantic alignment, and image retrieval accuracy, demonstrating strong generalization and cross-subject transferability under data-limited scenarios.

| Method | Voxel-Level | | | Semantic-Level (via decoding) | | | | Image Retrieval | |
|---|---|---|---|---|---|---|---|---|---|
| | MSE ↓ | Pearson ↑ | Cosine ↑ | Incep ↑ | CLIP ↑ | Eff ↓ | SwAV ↓ | Raw ↑ | Syn ↑ |
| LinearReg | .102 | .676 | .693 | 85.8% | 83.7% | .759 | .454 | – | – |
| GNet | .092 | .707 | .740 | 87.7% | 85.0% | .730 | .428 | – | – |
| SynBrain | **.079** | **.715** | **.769** | **96.7%** | **95.9%** | **.619** | **.350** | **94.7%** | **99.3%** |

Table 8: Subject-specific visual-to-fMRI synthesis baseline performance on Subject 1.

| Method (LinearReg) | Voxel-Level | | | Semantic-Level (via decoding) | | | | Image Retrieval | |
|---|---|---|---|---|---|---|---|---|---|
| | MSE ↓ | Pearson ↑ | Cosine ↑ | Incep ↑ | CLIP ↑ | Eff ↓ | SwAV ↓ | Raw ↑ | Syn ↑ |
| Sub1→Sub2 (1h) | .228 | .482 | .550 | 76.6% | 76.7% | .861 | .514 | – | – |
| Sub1→Sub5 (1h) | .281 | .616 | .696 | 80.5% | 80.1% | .835 | .488 | – | – |
| Sub1→Sub7 (1h) | .230 | .475 | .543 | 75.7% | 75.4% | .863 | .512 | – | – |

Table 9: Subject-adaptive visual-to-fMRI synthesis baseline performance using *LinearReg*.

# H  Additional Results

## H.1  Subject-specific Visual-to-fMRI Synthesis

To evaluate how SynBrain performs under subject-specific modeling settings, we independently train and evaluate SynBrain for four subjects (Sub1, Sub2, Sub5, Sub7). Table 10 reports quantitative results across voxel-level accuracy, semantic alignment, and cross-modal retrieval.

Overall, SynBrain achieves strong and consistent performance in all metrics across individuals. These results suggest that SynBrain can effectively model individual fMRI response distributions, producing neural representations that retain both structural fidelity and semantic consistency under subject-specific training.

## H.2  Few-shot Subject-adaptive Visual-to-fMRI Synthesis

To comprehensively assess the generalization ability of **SynBrain** under few-shot settings, we conduct *subject-adaptive visual-to-fMRI synthesis* by transferring models trained on one source subject to novel target subjects using only one hour of adaptation data. While the main paper reports transfers from Sub1 to Sub2, Sub5, and Sub7, here we extend the evaluation in two directions: (i) expanding to additional subjects and (ii) performing reverse-direction transfers.

**(i) Evaluation on Additional Subjects.** To examine SynBrain's ability to generalize across a more diverse population, we expanded our experiments to include all eight real subjects from the NSD dataset. The main experiments involved four subjects (Sub1, Sub2, Sub5, Sub7), whereas the remaining four (Sub3, Sub4, Sub6, Sub8), who completed fewer sessions, were incorporated for supplementary evaluation. As shown in Table 11 (Top), SynBrain maintains strong performance across these additional unseen individuals with only one hour of adaptation data, demonstrating robust generalization despite substantial inter-subject neural variability.

**(ii) Reverse-Direction Subject Adaptation.** We further investigated reverse transfer by adapting models pretrained on Sub2, Sub5, and Sub7 to Sub1. As presented in Table 11 (Bottom), all three source-subject models achieve consistently high semantic-level decoding accuracy (Inception: 88.0–88.9%, CLIP: 86.7–86.9%) and comparable voxel-level alignment after limited adaptation. Moreover, the fMRI-to-image retrieval accuracy based on the synthesized signals (Syn: 63.5–69.4%) closely matches the results observed in the Sub1→Others setting reported in the main text. These results confirm that SynBrain enables bidirectional cross-subject adaptation with minimal data, underscoring its robustness and adaptability in low-resource, cross-subject generalization scenarios.

## H.3  Data Augmentation for Few-shot fMRI-to-Image Decoding

We further evaluate the utility of SynBrain-generated fMRI signals as a data augmentation strategy for low-resource neural decoding. Beyond Sub1 (as shown in the main paper), we extend our experiments to Sub2, Sub5, and Sub7. For each target subject, we combine their 1-hour real fMRI data with synthetic fMRI signals generated by SynBrain (Sub1→SubX), using visual stimuli from a different,

| Method | Voxel-Level | | | Semantic-Level (via decoding) | | | | Image Retrieval | |
|---|---|---|---|---|---|---|---|---|---|
| | MSE ↓ | Pearson ↑ | Cosine ↑ | Incep ↑ | CLIP ↑ | Eff ↓ | SwAV ↓ | Raw ↑ | Syn ↑ |
| SynBrain (Sub1) | .079 | .715 | .769 | 96.7% | 95.9% | .619 | .350 | 94.7% | 99.3% |
| SynBrain (Sub2) | .134 | .667 | .715 | 96.2% | 94.5% | .637 | .364 | 92.4% | 97.1% |
| SynBrain (Sub5) | .190 | .739 | .793 | 96.2% | 93.9% | .627 | .356 | 78.0% | 86.4% |
| SynBrain (Sub7) | .151 | .626 | .680 | 93.6% | 93.0% | .671 | .380 | 73.9% | 87.3% |

Table 10: Quantitative subject-specific visual-to-fMRI synthesis performance.

| Method | Voxel-Level | | | Semantic-Level (via decoding) | | | | Image Retrieval | |
|---|---|---|---|---|---|---|---|---|---|
| | MSE ↓ | Pearson ↑ | Cosine ↑ | Incep ↑ | CLIP ↑ | Eff ↓ | SwAV ↓ | Raw ↑ | Syn ↑ |
| SynBrain (Sub1→Sub3, 1h) | .082 | .755 | .801 | 88.1% | 85.9% | .781 | .464 | 13.5% | 69.1% |
| SynBrain (Sub1→Sub4, 1h) | .089 | .739 | .798 | 89.0% | 86.9% | .759 | .455 | 13.5% | 73.1% |
| SynBrain (Sub1→Sub6, 1h) | .067 | .727 | .785 | 89.1% | 87.1% | .771 | .456 | 16.6% | 73.4% |
| SynBrain (Sub1→Sub8, 1h) | .146 | .620 | .687 | 85.1% | 82.8% | .807 | .486 | 8.9% | 69.9% |
| SynBrain (Sub2→Sub1, 1h) | .089 | .689 | .748 | 88.9% | 86.9% | .778 | .448 | 27.3% | 68.4% |
| SynBrain (Sub5→Sub1, 1h) | .086 | .691 | .752 | 88.1% | 86.7% | .780 | .446 | 20.7% | 63.5% |
| SynBrain (Sub7→Sub1, 1h) | .086 | .685 | .746 | 88.0% | 86.7% | .768 | .441 | 23.9% | 69.4% |

Table 11: Quantitative few-shot visual-to-fMRI synthesis performance on novel subjects.

unseen hour. This augmented dataset is used to train the fMRI-to-image decoder (*MindEye2+DA*), which is compared against the baseline *MindEye2* and state-of-the-art *MindAligner* trained only on 1 hour of real data.

As shown in Table 12, *MindEye2+DA* consistently outperforms MindEye2 across all subjects. Notable improvements are observed in high-level semantic metrics (e.g., Inception Score and CLIP) and image-to-brain retrieval accuracy, suggesting that the generated fMRI signals are more semantically aligned with the intended visual semantics. Moreover, *MindEye2+DA* even outperforms MindAligner [11] across all low-level and high-level metrics, reaching state-of-the-art performance in few-shot (1h) fMRI-to-image decoding tasks. However, we also note a slight drop in brain-to-image retrieval accuracy (i.e., retrieving ground-truth images given fMRI signals) after augmentation. We hypothesize that this is due to the semantic nature of the synthetic fMRI signals, lacking low-level perceptual details such as pose, background, or texture. While this semantic consistency benefits cross-modal alignment and robustness, it may reduce instance-specific discriminability.

Nevertheless, these results confirm that SynBrain-generated fMRI representations are sufficiently realistic and semantically aligned with visual stimuli to serve as effective training signals for downstream brain decoding models, particularly in few-shot regimes.

**Trade-off between Naturalistic and Synthetic Data.** Here we conduct additional experiments with extended synthetic data durations (up to 32 hours) and observed that the **optimal performance is achieved with just 1 hour of data augmentation**. As shown in Table 13, adding more synthetic data beyond this point leads to diminishing or even negative returns. This suggests a **trade-off between real and generated data**: while a small amount of high-quality synthetic data is beneficial, excessive augmentation may introduce distributional mismatch or redundancy, limiting further gains.

### H.4 Brain Functional Analysis

To further examine the biological plausibility of SynBrain's generated fMRI representations, we perform qualitative analysis of whole-brain activation patterns across subjects. Figures 8 and 9 visualize voxel-wise activation maps under both subject-specific and few-shot adaptation settings, respectively.

In Figure 8, we present four representative visual stimuli and compare the generated activation maps across four subjects (Sub1, Sub2, Sub5, Sub7). Despite individual variability in cortical anatomy and voxel organization, SynBrain produces consistent activation distributions across subjects in response to the same stimulus. The decoded images from different subjects also retain semantic consistency, suggesting that the synthesized neural patterns encode shared perceptual features.

Figure 9 further compares activation maps from SynBrain models trained under full-data and few-shot (1-hour) settings. For each stimulus category (airplane, face, giraffe, pizza), we visualize the

| Method | Low-Level | | | | High-Level | | | | Retrieval | |
|---|---|---|---|---|---|---|---|---|---|---|
| | PixCorr ↑ | SSIM ↑ | Alex-2 ↑ | Alex-5 ↑ | Incep ↑ | CLIP ↑ | Eff ↓ | SwAV ↓ | Image ↑ | Brain ↑ |
| MindEye2 (Sub2, 1h) | .200 | **.433** | 85.0% | 92.1% | 81.9% | 79.4% | .807 | .467 | 90.5% | 67.2% |
| MindAligner (Sub2, 1h) | .218 | .426 | 88.1% | 93.3% | 84.1% | 82.5% | .791 | .452 | 90.0% | 85.6% |
| **MindEye2+DA (Sub2, 1h)** | **.222** | .422 | **88.2%** | **94.0%** | **85.0%** | **83.0%** | **.781** | **.443** | 83.7% | 79.3% |
| MindEye2 (Sub5, 1h) | .175 | .405 | 83.1% | 91.0% | 84.3% | 82.5% | .781 | .444 | 66.9% | 47.0% |
| MindAligner (Sub5, 1h) | .197 | .409 | 84.7% | 91.6% | 84.6% | 82.8% | .784 | .454 | 70.6% | 66.0% |
| **MindEye2+DA (Sub5, 1h)** | **.200** | **.410** | **86.5%** | **93.0%** | **87.2%** | **86.1%** | **.753** | **.423** | 64.9% | 60.7% |
| MindEye2 (Sub7, 1h) | .170 | .408 | 80.7% | 85.9% | 74.9% | 74.3% | .854 | .504 | 64.4% | 37.8% |
| MindAligner (Sub7, 1h) | .183 | .407 | 81.5% | 88.3% | 79.9% | 77.8% | .834 | .487 | 64.2% | 62.6% |
| **MindEye2+DA (Sub7, 1h)** | **.189** | **.409** | **83.8%** | **90.1%** | **81.7%** | **79.3%** | **.809** | **.462** | 59.6% | 57.9% |

Table 12: Quantitative few-shot *fMRI-to-image decoding* performance comparisons on novel subjects.

| Method | Low-Level | | | | High-Level | | | | Retrieval | |
|---|---|---|---|---|---|---|---|---|---|---|
| | PixCorr ↑ | SSIM ↑ | Alex-2 ↑ | Alex-5 ↑ | Incep ↑ | CLIP ↑ | Eff ↓ | SwAV ↓ | Image ↑ | Brain ↑ |
| MindEye2(1h) | .235 | **.428** | 88.0% | 93.3% | 83.6% | 80.8% | .798 | .459 | **94.0%** | 77.6% |
| MindEye2(1h)+DA(1h) | **.243** | .419 | **90.1%** | **95.1%** | **85.1%** | **84.7%** | **.770** | **.432** | 87.9% | **82.0%** |
| MindEye2(1h)+DA(8h) | .222 | .417 | 88.5% | 93.6% | 84.9% | 82.5% | .795 | .450 | 67.1% | 68.8% |
| MindEye2(1h)+DA(16h) | .229 | .416 | 88.4% | 93.4% | 83.1% | 81.8% | .812 | .457 | 64.8% | 63.7% |
| MindEye2(1h)+DA(32h) | .222 | .410 | 88.4% | 93.1% | 82.8% | 81.8% | .813 | .457 | 61.6% | 61.9% |

Table 13: Few-shot *fMRI-to-image decoding* with various hours of synthetic data.

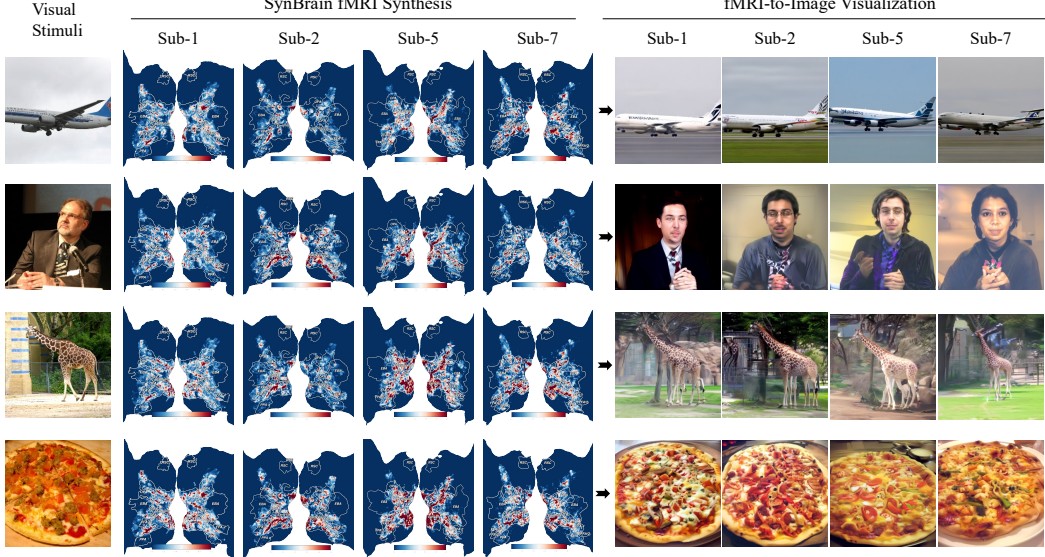

Figure 8: Comparisons of activation maps and fMRI-to-Image visualizations across subjects evoked by the same visual stimuli. All models are trained on full data (40h) from specific subjects.

generated activations from the fully trained model (i.e., Sub2, Sub5, Sub7) and the few-shot adapted model (i.e., Sub1→Sub2, Sub1→Sub5, Sub1→Sub7). The results show that even under limited data conditions, the few-shot models can preserve key category-specific activation patterns, such as enhanced responses in occipital and ventral temporal areas for high-level object categories.

These findings suggest that SynBrain not only achieves high-level semantic alignment but also captures spatially meaningful and category-sensitive neural patterns that are consistent across subjects under data-limited regimes.

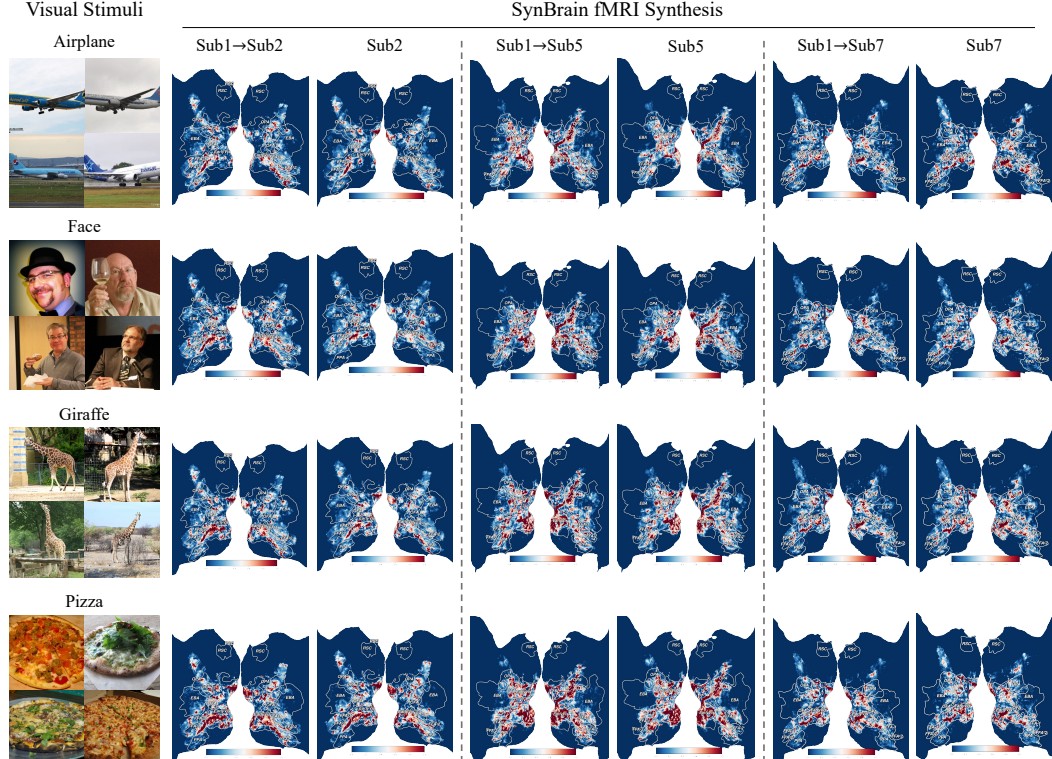

Figure 9: Comparisons of activation maps between full-data (40h) training (i.e., Sub2, Sub5, Sub7) and few-shot (1h) adaptation (i.e., Sub1→Sub2, Sub1→Sub5, Sub1→Sub7) across subjects evoked by representative categories of visual stimuli.

