# OpenReview forum: "SynBrain: Enhancing Visual-to-fMRI Synthesis via Probabilistic Representation Learning"
_NeurIPS.cc/2025/Conference — NeurIPS 2025 poster_

### Official Review · Reviewer_DDYY · 2025-06-23

**Clarity:** 3
**Significance:** 3
**Originality:** 3
**Rating:** 5
**Confidence:** 4

**Summary:**

This paper presents SynBrain, a generative model that learns to synthesize fMRI responses from visual input using a probabilistic approach. The key idea is to model brain responses as distributions rather than fixed outputs, better reflecting biological variability. The method performs well and shows good potential for few-shot adaptation.

**Questions:**

In BrainVAE, fMRI responses are modeled as a latent probability distribution z, capturing the one-to-many mapping. However, in the S2N Mapper, CLIP features are mapped deterministically to a latent feature, resembling a one-to-one mapping. Does this not contradict the biological argument that visual-to-neural mappings are inherently one-to-many? Could the authors clarify whether this deterministic mapping risks averaging out noise or inter-trial variability, and what impact this has on downstream synthesis quality and biological realism?

**Ethical Concerns:**

["NO or VERY MINOR ethics concerns only"]

**Final Justification:**

The authors have addressed my concerns in the rebuttal.

**Limitations:**

See weaknesses.

**Quality:**

3

**Strengths And Weaknesses:**

Strengths:
1. The use of a semantic-conditioned probabilistic framework to model the one-to-many mapping from visual stimuli to neural responses is both biologically plausible and methodologically sound. This explicitly accounts for trial-level noise, attentional fluctuations, and inter-subject variability.
2. The proposed method achieves state-of-the-art performance in visual-to-fMRI encoding across several settings.
3. The generated neural signals are shown to be biologically interpretable.

Weaknesses:
1. The main experiments are conducted on a single-subject setting. Given the increasing importance of multi-subject modeling in this field, it would strengthen the paper to explore whether incorporating data from multiple source subjects could further improve generalization. For instance, in Table 1, the few-shot adaptation experiments (1→2/5/7) could be extended to include a multi-subject pretraining setting (e.g., 125→7).
2. The few-shot adaptation experiments lack comparison to suitable baselines. For example, could MindSimulator be fine-tuned on a new subject in a similar few-shot setting, and how would it compare?
3. In Table 2, the benefit of data augmentation with MindEye2(1h)+DA(1/2/4h) is shown, but the upper bound of the benefit remains unclear. It would be helpful to analyze how much synthetic data is optimal and whether the performance saturates with more hours of augmented data.

---

> ### Author Rebuttal · Authors · 2025-07-31
>
> **W1.** Thank you for the thoughtful suggestion. We conducted 1-hour novel-subject adaptation experiments using multi-subject pretraining (e.g., Sub125→Sub7, Sub257→Sub1) and observed modest performance gains compared to single-subject pretraining (e.g., Sub1→Sub7, Sub2→Sub1), as shown below.
>
> However, these improvements required deeper Transformer architectures (8→12 layers) and extended training (50k→200k steps), substantially increasing computational cost. In contrast, single-subject pretraining followed by 1-hour adaptation offers a more efficient trade-off between performance and training resources.
>
> | SynBrain (1h-Adapt)    | MSE↓  | Pearson↑ | Cosine↑ | Incep↑ | CLIP↑ | Eff↓  | SwAV↓ | Raw↑  | Syn↑  |
> |--------------------|-------|----------|---------|--------|--------|--------|--------|--------|--------|
> | Sub1→Sub7          | 0.151 | 0.630    | 0.679   | 86.8%  | 84.7%  | 0.783 | 0.453 | 13.2% | 76.5% |
> | Sub125→Sub7        | 0.137 | 0.658    | 0.695   | 87.2%  | 85.0%  | 0.795 | 0.449 | 14.2% | 78.3% |
> | Sub2→Sub1          | 0.089 | 0.689    | 0.748   | 88.9%  | 86.9%  | 0.778 | 0.448 | 27.3% | 68.4% |
> | Sub257→Sub1        | 0.080 | 0.693    | 0.774   | 89.4%  | 87.3%  | 0.773 | 0.452 | 31.6% | 71.7% |
>
> **W2.** Thank you for the suggestion. Taking MindSimulator as a cross-subject adaptation baseline is limited by two practical constraints. i) **MindSimulator is not open-sourced** and incorporates several **hand-crafted** components (e.g., resting-state initialization, custom noise injection), making it **difficult to reproduce**; ii) MindSimulator’s Autoencoder is **subject-specific** and tightly coupled to **fixed voxel dimensions**, rendering it incompatible with cross-subject scenarios where ROI sizes vary. These limitations prevent a direct and meaningful comparison between SynBrain and MindSimulator in the context of cross-subject adaptation.
>
> To establish a transparent and reliable baseline, we implemented **Linear Regression**, the most classical encoding model, using a simple two-layer linear architecture. The first layer maps CLIP features to a 4,096-dimensional hidden space, and the second layer projects to subject-specific voxel responses. During few-shot adaptation, we freeze the first layer and train only the second layer using 1-hour data from the target subject. As shown in the results below, **SynBrain consistently outperforms Linear Regression** across voxel-wise correlation, semantic alignment, and image retrieval accuracy, demonstrating strong generalization and cross-subject transferability under data-limited scenarios.
>
> | Method           | 1h-Adapt | MSE↓ | Pearson↑ | Cosine↑ | Incep↑ | CLIP↑ | Eff↓ | SwAV↓ | Raw↑ | Syn↑ |
> |------------------|------------|--------|------------|-----------|-----------|---------|--------|----------|--------|--------|
> | SynBrain         | Sub1→Sub2  | 0.160  | 0.619      | 0.675     | 89.2%    | 88.1%  | 0.751  | 0.431    | 19.5%  | 67.4%  |
> |                  | Sub1→Sub5  | 0.224  | 0.704      | 0.765     | 89.2%    | 88.0%  | 0.749  | 0.432    | 16.9%  | 54.8%  |
> |                  | Sub1→Sub7  | 0.151  | 0.630      | 0.679     | 86.8%    | 84.7%  | 0.784  | 0.453    | 13.2%  | 76.5%  |
> | LinearReg| Sub1→Sub2  | 0.228  | 0.482      | 0.550     | 76.6%    | 76.7%  | 0.861  | 0.514    |   –    |   –    |
> |                  | Sub1→Sub5  | 0.281  | 0.616      | 0.696     | 80.5%    | 80.1%  | 0.835  | 0.488    |   –    |   –    |
> |                  | Sub1→Sub7  | 0.230  | 0.475      | 0.543     | 75.7%    | 75.4%  | 0.863  | 0.512    |   –    |   –    |
>
>
> **W3.** Thank you for the insightful question. We conducted additional experiments with extended synthetic data durations (up to 32 hours) and observed that the **optimal performance is achieved with just 1 hour of data augmentation**. As shown below, adding more synthetic data beyond this point leads to diminishing or even negative returns.
>
> This suggests a **trade-off between real and generated data**: while a small amount of high-quality synthetic data is beneficial, excessive augmentation may introduce distributional mismatch or redundancy, limiting further gains.
>
> | Method                 | PixCorr↑ | SSIM↑ | Alex2↑ | Alex5↑ | Incep↑ | CLIP↑ | Eff↓ | SwAV↓ | Image↑ | Brain↑ |
> |------------------------|-----------|--------|---------|---------|----------|---------|--------|---------|----------|----------|
> | MindEye2(1h)    | 0.235     | **0.428**  | 88.0%   | 93.3%   | 83.6%    | 80.8%   | 0.798  | 0.459   | **94.0%**    | 77.6%    |
> | MindEye2(1h)+DA(1h)    | **0.243**     | 0.419  | **90.1%**   | **95.1%**   | **85.1%**    | **84.7%**   | **0.770**  | **0.432**   | 87.9%    | **82.0%**    |
> | MindEye2(1h)+DA(8h)    | 0.222     | 0.417  | 88.5%   | 93.6%   | 84.9%    | 82.5%   | 0.795  | 0.450   | 67.1%    | 68.8%    |
> | MindEye2(1h)+DA(16h)   | 0.229     | 0.416  | 88.4%   | 93.4%   | 83.1%    | 81.8%   | 0.812  | 0.457   | 64.8%    | 63.7%    |
> | MindEye2(1h)+DA(32h)   | 0.222     | 0.410  | 88.4%   | 93.1%   | 82.8%    | 81.8%   | 0.813  | 0.457   | 61.6%    | 61.9%    |
>
> **Q1.** We thank the reviewer for this insightful question. While the S2N Mapper adopts a deterministic mapping from CLIP features to BrainVAE’s latent space, this design does not contradict the biological one-to-many nature of visual-to-neural mappings.
> In the **first stage**, BrainVAE is trained to capture trial-level variability by associating each visual input with a latent distribution, **whose center captures the most salient and semantically consistent neural features across trials**. By conditioning on this center and performing **stochastic sampling**, BrainVAE can generate multiple plausible fMRI responses that are semantically aligned yet voxel-wise diverse, faithfully reflecting the one-to-many nature of neural encoding.
>
> In the **second stage**, the S2N Mapper is optimized to align each CLIP embedding with this **semantic anchor**—the **center of the latent distribution**—rather than with any specific trial or empirical mean. Its objective is to map visual features to the **most representative position** in latent space, while **delegating the modeling of trial-level variability to the stochastic sampling process**. This design ensures that variability is not averaged out, but reintroduced in a controlled, biologically grounded manner.
>
> As shown in **Supplementary Figure 2**, repeated synthesis from the same visual input yields fMRI signals that are **voxel-wise diverse but semantically stable**. This confirms that our framework supports semantic-consistent sampling and generation, effectively balancing biological realism with representational precision.
>
> We thank the reviewer again for the valuable feedback. We hope our response adequately addresses your comments. If so, we hope you could consider a more positive evaluation of our work. If there are any remaining concerns, we would be happy to further address them during the discussion phase.

---

> > ### Comment · Reviewer_DDYY · 2025-08-05
> >
> > Thanks for the detailed reply. The authors’ clarifications were clear and addressed all of my concerns. I have updated my score to 5.

---

> > > ### Author Response · Authors · 2025-08-08
> > >
> > > Thank you for your positive and thoughtful feedback. We sincerely appreciate the time and effort you devoted to reviewing our work and offering valuable comments.

---

### Official Review · Reviewer_fiTc · 2025-06-23

**Clarity:** 3
**Significance:** 3
**Originality:** 3
**Rating:** 5
**Confidence:** 4

**Summary:**

This paper proposes SynBrain, a generative framework for visual-to-fMRI synthesis that models neural responses as semantic-conditioned probability distributions. The key innovation lies in addressing the inherent one-to-many mapping from visual stimuli to neural responses through probabilistic modeling. SynBrain consists of two main components: (1) BrainVAE, a variational autoencoder that models fMRI signals as continuous probability distributions conditioned on visual semantics, and (2) S2N (Semantic-to-Neural) Mapper, a Transformer-based module that maps CLIP visual embeddings to the fMRI latent space. The authors evaluate their approach on the Natural Scenes Dataset (NSD) with 4 subjects, demonstrating superior performance in subject-specific visual-to-fMRI synthesis and effective few-shot adaptation to novel subjects.

**Questions:**

1. How does the computational cost of SynBrain compare to deterministic baselines? The paper mentions faster training but doesn't provide detailed timing comparisons or discussions. Is this impacted by the necessary computing hardware to train and run their method? The authors note that they use 4 A100's to train their method, it is possible to train it on a single GPU or with reduced hardware requirements? Does  the variational approach have meaningful differences in inference efficiency?

2. The authors claim that synthesized fMRI signals achieve higher retrieval accuracy (92.5%) than raw fMRI (84.8%). Could this indicate that the model is learning to enhance or denoise the signals rather than faithfully reproduce the noisy properties of neural responses? It seems to me that one of the primary advantages of a variational system is the ability to more accurately model the noise properties of real neural data, what do the authors make of this point in the context of the "better" retrieval scores?

3. How sensitive is the approach to the choice of visual encoder (CLIP)? Have the authors experimented with other pre-trained vision models?

4. The few-shot adaptation experiments use subjects from the same dataset. How would the approach perform when adapting to subjects scanned with different protocols or equipment? Transferring to new subjects with minimal data in new datasets would be of significant interest to downstream applications of the approach and would boot the significance of the work.

5. Could the authors provide more analysis on what specific types of neural variability the model captures versus what it might be missing?

6. On line 171 the authors note that MindEye uses a diffusion transformer, while the MindEye paper actually uses a *diffusion prior*. This detail should be corrected for the camera ready version.

**Ethical Concerns:**

["NO or VERY MINOR ethics concerns only"]

**Final Justification:**

I am impressed by the depth of their rebuttal, and consider the majority of my concerns addressed. In particular, the drastic improvements fine tuning across datasets, the massive boost in inference efficiency, additional baselines, and increased clarity into the training efficiency of their model I think are all important items that boost the significance of their submission. I will increase my score to 5, as I believe the paper represents a high-impact improvement to the paradigm and performance of visual encoding models for fMRI.

**Limitations:**

The authors adequately address limitations in the conclusion, including reliance on pre-trained vision models and incomplete modeling of variance sources. However, they could expand on computational requirements, unknown dataset transferability, and any scalability concerns.

**Quality:**

3

**Strengths And Weaknesses:**

**Strengths:**

**Quality:**
- The paper addresses a fundamental challenge in computational neuroscience by explicitly modeling the one-to-many nature of visual-to-neural mappings, which is biologically motivated and theoretically sound.
- The experimental design is comprehensive, evaluating across voxel-level, semantic-level, and retrieval metrics, providing multi-faceted validation of the approach.
- The ablation studies clearly demonstrate the contribution of each component (variational sampling, contrastive learning, S2N Mapper).
- The architecture design is well-motivated, particularly the use of convolutional and attention layers in BrainVAE to address instability issues in MLP-based VAEs.

**Clarity:**
- The paper is generally well-written with clear motivation and methodology sections.
- The figures effectively illustrate the approach and results, particularly Figure 2 showing the overall framework.
- The experimental setup is clearly described with sufficient implementation details for reproducibility.

**Significance:**
- The work addresses an important limitation in existing deterministic approaches to brain encoding, offering a more biologically plausible framework.
- The results demonstrate practical benefits for data augmentation in few-shot fMRI-to-image decoding scenarios.
- The findings about cross-trial and cross-subject functional consistency provide insights into neural variability patterns.

**Originality:**
- The combination of probabilistic modeling with semantic conditioning for fMRI synthesis is novel and well-motivated.
- The S2N Mapper design offers a more efficient alternative to diffusion-based approaches used in prior work.
- The framework successfully balances modeling neural variability while preserving functional consistency.

**Weaknesses:**

**Quality:**
- The evaluation is limited to a single dataset (NSD). While this is the largest and highest quality dataset available, validation on additional datasets would strengthen the claims, and I am particulary interested to see how the variational approach proposed would handle lower quality and smaller datasets.
- The comparison baseline is primarily limited to MindSimulator. The paper would benefit from comparisons to other established image-to-fMRI encoding models, such as the GNet model introduced in "Brain-optimized deep neural network models of human visual areas learn non-hierarchical representations" (St-Yves et al., 2023) and the Transformer-based approach from "Predicting Human Brain States with Transformer" (Sun et al., 2024). These comparisons would better contextualize SynBrain's performance relative to the broader landscape of visual encoding models, even if these baselines are deterministic.

**Clarity:**
- Some technical details about the BrainVAE architecture could be clearer, particularly the specific design choices and architectural details for the convolutional and attention components. This could have been bolstered by making code accessible to reviewers as well.

**Significance:**
- While the results are promising, the practical impact may be limited by the computational overhead of the probabilistic approach compared to deterministic methods. Can the authors comment on the computational efficiency of the method compared to existing approaches?
- The few-shot adaptation results, while encouraging, are evaluated on relatively similar subjects from the same dataset and the same scanner. Without cross-dataset adaptation results it is difficult to quantify the true impact of this result.

**Originality:**
- While the overall approach is novel, individual components (VAEs, contrastive learning, Transformers) are well-established techniques.
- The novelty lies primarily in their combination and application to fMRI synthesis rather than fundamental algorithmic innovations.

---

> ### Author Rebuttal · Authors · 2025-07-31
>
> **W-Clarity1.** Thanks for the feedback. We will clarify the architectural details of BrainVAE in the revised manuscript. While the code cannot be shared during rebuttal for anonymity, it will be released upon acceptance to ensure reproducibility.
>
> **W-Quality2.** Thank you for the suggestion. Note that Predicting Human Brain States with Transformer (Sun et al., 2024) focuses on resting-state fMRI and aims to predict future brain states based on past activity. As such, it does not fall within the scope of **visual-to-fMRI encoding** and is not directly comparable to our setting. Hence, we have included additional comparisons with two representative baselines: (1) **GNet** (St-Yves et al., 2023), and (2) **Linear Regression**, a classical deterministic voxel-wise visual encoding model, using a simple two-layer linear architecture with a 4,096-dimensional hidden space.
>
> As shown below, SynBrain achieves consistently **superior results across multiple levels**, highlighting its effectiveness in fMRI synthesis with both voxel-level structure and semantic consistency.
>
> | Method     | MSE↓  | Pearson↑ | Cosine↑ | Incep↑ | CLIP↑ | Eff↓  | SwAV↓ | Raw↑  | Syn↑  |
> |------------|--------|-----------|-----------|----------|---------|--------|---------|--------|--------|
> | SynBrain   | **0.079**  | **0.715** | **0.769** | **96.7%** | **95.9%** | **0.619** | **0.350** | **94.7%** | **99.3%** |
> | LinearReg  | 0.102  | 0.676     | 0.693     | 85.8%   | 83.7%   | 0.759  | 0.454   | –      | –      |
> | GNet       | 0.092 | 0.707     | 0.740     | 87.7%   | 85.0%   | 0.730  | 0.428   | –      | –      |
>
> **Q1.**
>
> **Training Efficiency.**
> The faster training of BrainVAE is mainly due to its **rapid convergence** within fewer epochs. Under the **same total batch size** of 24, BrainVAE completes 15 epochs in ~2 hours (4×A100-40G), while MLP-AE requires 50 epochs and ~5 hours (1×A100-40G), yet still underperforms BrainVAE across key performance metrics.
> Note that under the same total batch size, multi-GPU training introduces extra overhead due to gradient synchronization across devices, resulting in a slightly longer per-epoch time compared to single-GPU training. Nonetheless, BrainVAE significantly shortens the overall training time by converging within far fewer epochs.
>
> **Hardware Flexibility.**
> While the results reported in the main paper use 4×A100-40G GPUs (with downsampling blocks = 1, latent dim = 4096), we also experimented with **lighter variants** of BrainVAE, as reported in **Supplementary Table 1**. For example, setting the number of **downsampling blocks to 3 (latent dim = 1024)** enables faster training on a **single** A100-40G GPU, with only minor performance degradation (-1.5% in Incep and -1.4% in CLIP). Even so, it still significantly outperforms MindSimulator (5-Trial) by 2.4% (Incep) and 4.7% (CLIP), demonstrating the scalability of SynBrain to more constrained computational environments.
>
> **Inference Efficiency.**
> The observed difference in inference efficiency is not inherently due to the variational approach itself, but rather stems from the design of architecture used in the Visual-to-fMRI synthesis process. Specifically, MindSimulator employs a multi-step diffusion-based method (Diffusion Transformer, DiT), while SynBrain uses **a one-step S2N mapper for direct sampling**. This leads to a substantial difference in inference time as shown below.
>
> | Method              | Inference Time(s) |
> |:------------------- |:-----------------:|
> | SynBrain (S2N)      |       0.059       |
> | MindSimulator (DiT) |      31.845       |
>
> **Q2.** **Semantic Consistency vs. Voxel-level Fidelity**
>
> Thanks for your comment. While BrainVAE is capable of modeling the noise characteristics inherent in neural signals through its latent space, it is **not intended to faithfully reconstruct the full voxel-level complexity** of raw fMRI signals. Instead, by integrating visual semantic constraints during training, the model **prioritizes semantic consistency over voxel-wise fidelity**, resulting in synthesized signals that retain relevant variability while **suppressing irrelevant noise**.
>
> The improved retrieval accuracy reflects this design choice. Rather than contradicting the variational framework, it highlights the capability of BrainVAE to **balance biologically plausible variability with task-relevant constraints**, enabling effective sampling and generation aligned with perceptual semantics. As shown in **Supplementary Figure 2**, fMRI synthesis via multiple stochastic passes of BrainVAE exhibit voxel-level variability, yet remain **semantically consistent across samples**.
>
>
> **Q3.** **Vision Model Comparison:**
>
> Thanks for your thoughtful question. We evaluated the impact of visual encoder choice by replacing CLIP in SynBrain with two advance visual foundation models, **DINOv2** and **MAE**. As shown in the table below, CLIP achieves the best overall performance across reconstruction, voxel-level correlation, and semantic alignment. We attribute this to its vision-language contrastive objective, which provides stronger semantic supervision compared to purely visual models like DINOv2 and MAE. Nonetheless, the relatively strong performance of these alternatives indicates that SynBrain remains robust across different encoder choices.
>
> We also include a comparison with MindSimulator (5-Trial), which likewise uses CLIP as its visual encoder. Notably, MindSimulator averages results over five generated fMRI trials to improve performance, whereas SynBrain achieves superior results with just a single trial. This suggests that SynBrain benefits not only from a strong encoder but also from more effective model design and training strategy.
>
> | Method                  | VisEncoder | MSE↓  | Pearson↑ | Cosine↑ | Incep↑ | CLIP↑  | Eff↓  | SwAV↓ | Raw↑  | Syn↑  |
> |-------------------------|----------------|-------|----------|---------|--------|--------|-------|--------|-------|-------|
> | MindSimulator (5-Trial) | CLIP           | 0.417 | 0.326    | -       | 92.8%  | 89.8%  | 0.714 | 0.402  | -     | -     |
> | SynBrain (1-Trial)      | CLIP           | 0.079 | 0.715    | 0.769   | 96.7%  | 95.9%  | 0.619 | 0.350  | 94.7% | 99.3% |
> |                         | DINOv2         | 0.089 | 0.679    | 0.728   | 92.6%  | 91.8%  | 0.701 | 0.375  | 76.3% | 87.6% |
> |                         | MAE            | 0.097 | 0.676    | 0.737   | 91.0%  | 90.5%  | 0.705 | 0.389  | 68.6% | 83.7% |
>
>
> **W1&Q4.** **Cross-dataset Adaptation**
>
> Thanks for the suggestions. To tackle this concern, we conducted extra experiments on the GOD [1] dataset, which differs from NSD in both scanning protocol and data quality. Each GOD subject contains only 1,200 training images collected in a single session and 50 out-of-class test images, offering a challenging setting for cross-dataset adaptation.
>
> GOD subjects used the Visual ROI defined by MindVis (4,623 voxels), and for consistency, we employed a SynBrain variant trained on the NSD Visual ROI (4,657 voxels), instead of the standard NSDGeneral ROI (15,724 voxels, from MindEye). This variant features 3 downsampling blocks ($dim=512$) that enable single-GPU training.
>
> We compared two conditions on GOD: (1) training SynBrain from scratch, and (2) adapting a SynBrain model pretrained on NSD (Sub1). As shown in the table below, the **pretrained model substantially outperforms the from-scratch baseline** across all metrics.
>
> Note that, in this case, **voxel-level correlations remain relatively low**—primarily due to the **poor SNR** (Signal-Noise Ratio) of GOD (~0.1, as reported in MindVis [2]). However, the model achieves **substantial gains at the semantic level**. This suggests that under such noisy conditions, high voxel-wise correlation may reflect overfitting to noise rather than capturing meaningful neural structure. SynBrain, by contrast, prioritizes semantic alignment and achieves $68.0$% **50-way Top-1 retrieval accuracy** using synthesized fMRI, outperforming Mind-Vis (pretrained on HCP), which reaches 23.9% with raw fMRI. These results highlight SynBrain’s effectiveness in transferring to small-scale, low-SNR datasets.
>
>
> | Method         | Pretrained   | MSE↓   | Pearson↑ | Cosine↑ | Raw↑  | Syn↑  |
> |----------------|--------------|---------|-----------|-----------|--------|--------|
> | Mind-Vis       | HCP          | –       | –         | –         | 23.9%  | –      |
> | SynBrain-GOD   | –            | 18.283  | 0.104     | 0.119     | 6.0%   | 12.0%  |
> | SynBrain-GOD   | NSD(Sub1)   | 6.845   | 0.151     | 0.176     | 13.0%  | 68.0%  |
>
>
> [1] Horikawa, T. & Kamitani, Y. Generic decoding of seen and imagined objects using hierarchical visual features. Nature Communications 2017.
>
> [2] Chen Z, et al. Seeing beyond the brain: Conditional diffusion model with sparse masked modeling for vision decoding. CVPR 2023.
>
> **Q5.** Thank you for the valuable question. BrainVAE is designed to capture **stimulus-driven, semantically relevant variability** by modeling **trial-level one-to-many mappings**—that is, the distribution of neural responses elicited by repeated or similar visual stimuli. This allows the model to learn not just a single canonical response per stimulus, but a functionally consistent response space shaped by perceptual semantics. While the model can, in principle, capture broader sources of variability, **subject-specific idiosyncrasies and spontaneous background activity** are typically down-weighted, as they are less aligned with visual semantics.
>
> **Q6.** Thank you for pointing this out. We acknowledge the terminology difference and will revise the description accordingly in the camera-ready version.
>
> We thank the reviewer again for the valuable feedback. We hope our response adequately addresses your comments. If so, we hope you could consider a more positive evaluation of our work. If there are any remaining concerns, we would be happy to further address them during the discussion phase.

---

> > ### Comment · Reviewer_fiTc · 2025-08-02
> >
> > Thanks to the authors for addressing my concerns and questions. I am impressed by the depth of their rebuttal, and consider the majority of my concerns addressed. In particular, the drastic improvements fine tuning across datasets, the massive boost in inference efficiency, additional baselines, and increased clarity into the training efficiency of their model I think are all important items that boost the significance of their submission. I will increase my score to 5, as I believe the paper represents a high-impact improvement to the paradigm and performance of visual encoding models for fMRI.

---

> > > ### Author Response · Authors · 2025-08-08
> > >
> > > Thank you for your positive and thoughtful feedback. We sincerely appreciate the time and effort you devoted to reviewing our work and offering valuable comments.

---

### Official Review · Reviewer_sdRq · 2025-06-30

**Clarity:** 2
**Significance:** 4
**Originality:** 3
**Rating:** 4
**Confidence:** 4

**Summary:**

This paper introduces SynBrain, a generative framework that operates under the brain decoding scenario. Experiments show that SynBrain generates more coherent data compared to the baseline and can improve existing decoding models as a data augmentation engine under limited-data scenarios.

**Questions:**

- What kind of fMRI input was used? The "nsdgeneral" region, like previous works such as MindEye
- For the data augmentation experiments in table 2, did the authors try traditional data augmentation strategies as a comparison?
- Experimental details for figure 3:
    - Are the experiments about just swapping the brain encoder/decoder from the pipeline in figure 2?
    - Is the MSE loss evaluating only the autoencoder, while the other losses evaluate the synthesis pipeline?
- I'm not sure if it's a good idea to introduce two (possibly competing) objectives that directly optimize the latent z in equation 4. Is it really necessary to introduce the SoftCLIP loss to the latent, instead of some other part of the pipeline, when it is already conditioned by the KL loss? I understand that outright removing the SoftCLIP loss is harmful, as seen in table 3, but can't it be introduced in some other part instead of being directly fed into the latent z that's supposed to follow a Gaussian distribution?
- On a related note, how do the KL loss and SoftCLIP loss behave during training? Do $\lambda_{KL}$ and $\lambda_{CLIP}$ keep the two losses in the general ballpark during training? Do the latent vectors of the training samples properly follow a Gaussian distribution after training?

**Ethical Concerns:**

["NO or VERY MINOR ethics concerns only"]

**Final Justification:**

They have answered most of my questions and concerns. Many of them stem from the confusion I had while reading the manuscript, but I expect the authors to revise the manuscript to reflect my questions and confusion in the final version.

**Limitations:**

yes

**Quality:**

3

**Strengths And Weaknesses:**

## Strengths
- The proposed data synthesis pipeline outperforms the shown baselines.
- The proposed method can also be utilized to improve existing decoding models as a data augmentation strategy under limited-data scenarios.

## Weaknesses
- There are some concerns regarding the theoretical motivation of the proposed loss function (Eq. 4); more on this in the questions section.
- Details on some of the experiments, mainly figure 3, seem to be absent.

---

> ### Author Rebuttal · Authors · 2025-07-31
>
> **Q1.** Yes, we used the “nsdgeneral” cortical region as fMRI input, following previous works like MindEye.
>
>
> **Q2.** Thank you for the insightful question. In our experiments, the fMRI data generated by SynBrain was used as an additional augmentation strategy to further enhance the performance of MindEye2 in the 1-hour few-shot adaptation setting. It’s worth noting that MindEye2 already incorporates standard data augmentation techniques (i.e., image distortions and MixUp) during training. **Our intention was not to compare SynBrain with traditional augmentation methods, but rather to build upon them.** Specifically, we aimed to validate that SynBrain-generated fMRI can effectively complement real fMRI data, helping to alleviate the challenge of data scarcity and improve model generalization in low-resource scenarios.
>
> **Q3.** Thank you for the thoughtful questions, and we apologize for the lack of clarity in the current version of Figure 3. To clarify, **Figure 3 corresponds exclusively to Stage-1 (BrainVAE) in Figure 2**, which focuses on training the BrainVAE module. The experiments in Figure 3 investigate **the effect of replacing the encoder and decoder architecture within Stage-1**. Specifically, we compare our proposed BrainVAE with two alternative designs—MLP-AE and MLP-VAE—while keeping the loss functions and training setup identical. As defined in **Equation (4)**, all models are optimized using **KL loss, MSE loss, and CLIP loss**. The only difference lies in the encoder-decoder architecture.
>
> Regarding the second question, all loss terms are used to evaluate the Stage-1 (BrainVAE) only. The MSE loss measures **voxel-level reconstruction quality**, while the CLIP loss evaluates the **semantic consistency between the brain latent $z$ and the image embedding $z_{CLIP}$**. Figure 3 (Right) shows the validation loss curves (MSE and CLIP) during Stage-1 training for each architecture. To further assess the quality of the brain latent space, we also conduct **Brain-Image Retrieval between $z$ and $z_{CLIP}$** in Stage-1. Higher retrieval accuracy reflects stronger semantic alignment.
>
> Finally, we emphasize that **Figure 3 is entirely unrelated to Stage-2 (S2N Mapper) in Figure 2**; the experiments and comparisons are **solely focused on Stage-1 (BrainVAE)**.
>
> **Q4.** Thank you for your thoughtful and detailed question. In our design, the KL loss and SoftCLIP loss are applied to the latent variable z with **complementary purposes**: the KL loss encourages smoothness and sampleability in the latent space, while the SoftCLIP loss promotes semantic alignment between brain and image modalities.
>
> Importantly, we intentionally **downweight the KL loss**, allowing it to function primarily as a **soft regularizer** rather than strictly enforcing a standard Gaussian prior. This **relaxed constraint** gives the latent space flexibility to encode semantically aligned structure, while still preserving the stochastic sampling capability that underlies variational generation.
>
> As shown in **Table 3**, removing either KL (Variation Sampling) or CLIP (Contrastive Learning) objective significantly degrades performance, underscoring their necessity. Furthermore, **Supplementary Figure 2** demonstrates that their combination enables **semantically consistent sampling** from the latent space—achieving the one-to-many generation behavior that BrainVAE is designed to support.
>
> This design choice is also supported by prior work such as **VAVAE** [1], which demonstrated the effectiveness of combining semantic constraints from visual encoders (e.g., CLIP, DINOv2) with variational modeling.
>
> [1] Yao, J., Yang, B., & Wang, X. Reconstruction vs. generation: Taming optimization dilemma in latent diffusion models. CVPR 2025 Oral.
>
> **Q5.** Thank you for the insightful follow-up. As shown in Figure 3, both the MSE loss and CLIP loss decrease steadily over the first 15 epochs and then stabilize, indicating that the model converges in terms of both voxel-level reconstruction and semantic alignment. The KL loss follows a similar trajectory, converging within 15 epochs.
>
> To balance the three objectives, we apply appropriate weighting: $\lambda_{KL}=0.001$, $\lambda_{CLIP} = 1000$. After weighting, the KL loss remains at the order of $10^1$, MSE loss decreases from $10^3$ to $10^2$ over training, and CLIP loss stays around the order of $10^3$. This setup ensures that each loss term contributes in a controlled and complementary manner throughout training. In particular, the CLIP loss plays a more influential role in the early stages, helping guide the latent space toward semantic alignment, while the KL loss provides soft regularization to support sampling.
>
> Importantly, our goal is not for the latent vectors to strictly follow a standard Gaussian distribution, but to remain sufficiently regularized to support stochastic sampling. As further illustrated in **Supplementary Figure 2**, SynBrain can perform stochastic sampling in the latent space (via noise injection) and still generate semantically consistent fMRI signals. This capability reflects the core design principle of BrainVAE: to **balance variational sampling with meaningful semantic structure**, rather than enforce a rigid Gaussian prior.
>
> We thank the reviewer again for the valuable feedback. We hope our response adequately addresses your comments. If so, we hope you could consider a more positive evaluation of our work. If there are any remaining concerns, we would be happy to further address them during the discussion phase.

---

> > ### Comment · Reviewer_sdRq · 2025-08-05
> >
> > I thank the authors for their response. They have answered most of my questions and concerns, and I have adjusted my score to a 4.

---

> > > ### Author Response · Authors · 2025-08-08
> > >
> > > Thank you for your positive and thoughtful feedback. We sincerely appreciate the time and effort you devoted to reviewing our work and offering valuable comments.

---

### Official Review · Reviewer_45Ff · 2025-07-03

**Clarity:** 4
**Significance:** 3
**Originality:** 3
**Rating:** 5
**Confidence:** 4

**Summary:**

This current model addresses a one-to-many relationship in stimulus to brain mapping by transforming semantically rich visual stimuli into synthesized neural activity patterns. The authors introduce a non-deterministic model in which both seen and unseen visual stimuli are used to generate synthesized fMRI data. The model performance is reported through a series of comparisons between raw and synthetic voxel activity as well as the resulting generated images' similarity to the original images.

**Questions:**

Have you considered using a simulation to generate a dataset with more subjects? It is a bit hard to state that the model tested variability across humans (rather than within individuals) with only 4 subjects.

fMRI-to-Image vis -- some of the images generated from the synthesized fMRI capture the gist of the input visual stimulus but have very odd non-sensical distortions to them (e.g. the teddy bear has too many limbs; background vehicles in bus image), how would you explain these distortions? The model is supposed to capture high-level visual representations but seems like it is not reproducing semantic features in the background (vs the foreground).

**Ethical Concerns:**

["NO or VERY MINOR ethics concerns only"]

**Limitations:**

Yes

**Quality:**

3

**Strengths And Weaknesses:**

### Pros:

* The current model addresses high variability across humans (and thus across differences in individual visual systems), accounting for variance across physiological and trial-level constraints.

* They employ a novel non-linear transformation of CLIP embeddings into fMRI latent space.

* All together, the model is built to generate fMRI signals that represent high-level visual processing regions.

### Cons:

* (Very, very minor comment) please correct the typo in Figure 3 (Image Retrieval Accuracy).

* The dataset on which the model was tested had very few subjects. The Natural Scenes Dataset only contains 8 subjects in total (albeit several hours of viewing per subject), of which this paper could only utilize 4 subjects due to experiment completion. Please consider a simulation testing variability across more simulated individuals.

* Visual images generated (Figure 4) seem to capture the gist of the input visual stimulus but have some strange distortions.

---

> ### Author Rebuttal · Authors · 2025-07-31
>
> **W1.** Thanks for your correction.
>
> **W2 & Q1.** Thank you for the suggestion. While simulation-based evaluation is a promising direction, we chose to leave it to future work due to concerns about its **biological validity**. Most simulation methods rely on injecting noise or applying spatial transformations to existing fMRI data, but it remains unclear whether such approaches can faithfully replicate the **complex, subject-specific variability** observed in real neural responses.
>
> As shown in **Supplementary Figure 3**, even when presented with the same visual stimulus, **different subjects exhibit substantial differences in activation patterns**. This raises concerns about whether current simulation techniques can meaningfully benefit cross-subject modeling without introducing unrealistic assumptions.
>
> In line with the reviewer’s suggestion to evaluate on a more diverse population, we expanded our experiments to include **all eight real subjects** from the NSD dataset. Four subjects (Sub-1,2,5,7) were used in the main experiments, and the remaining four (Sub-3,4,6,8), who completed fewer sessions, were incorporated as supplementary evaluations to further assess SynBrain’s cross-subject few-shot adaptation capability.
>
> As shown below, SynBrain achieves strong performance across these unseen individuals with only 1 hour of adaptation data, demonstrating **robust generalization across real, subject-specific neural variability**.
>
> | SynBrain(1h)         | MSE↓ | Pearson↑ | Cosine↑ | Incep↑ | CLIP↑ | Eff↓ | SwAV↓ | Raw↑  | Syn↑  |
> |----------------|-------|------------|-----------|----------|---------|--------|--------|--------|--------|
> | Sub1→Sub3    | 0.082 | 0.755      | 0.801     | 88.1%    | 85.9%   | 0.781  | 0.464  | 13.5%  | 69.1%  |
> | Sub1→Sub4    | 0.089 | 0.739      | 0.798     | 89.0%    | 86.9%   | 0.759  | 0.455  | 13.5%  | 73.1%  |
> | Sub1→Sub6    | 0.067 | 0.727     | 0.785     | 89.1%    | 87.1%   | 0.771  | 0.456  | 16.6%  | 73.4%  |
> | Sub1→Sub8    | 0.146 | 0.620      | 0.687     | 85.1%    | 82.8%   | 0.807  | 0.486  | 8.9%  | 69.9%  |
> | Sub1→Sub2   | 0.160   | 0.619   | 0.675  | 89.2% | 88.1% | 0.751 | 0.431 | 19.5% | 67.4%  |
> | Sub1→Sub5   | 0.224   | 0.704   | 0.765  | 89.2% | 88.0% | 0.749 | 0.432 | 16.8% | 54.8%  |
> | Sub1→Sub7   | 0.151   | 0.630   | 0.679  | 86.8% | 84.7% | 0.783 | 0.453 | 13.2% | 76.5%  |
>
>
> **W3 & Q2.** Thanks for your comment. We agree that some images generated from the synthesized fMRI exhibit noticeable distortions, particularly in background details. These artifacts primarily stem from the limited capacity of the fMRI-to-image decoder used in our pipeline—namely, an **unrefined version of MindEye2** [1]. As shown in the original MindEye2 paper, even with ground-truth fMRI input, the reconstructed images can retain overall semantic content while exhibiting unnatural elements. This issue could be mitigated by replacing the decoder with a **refined version of MindEye2** (i.e., by guiding the unrefined reconstruction through base SDXL with image-to-image generation), which offers improved perceptual quality and semantic alignment.
>
> In our work, however, the **decoder** is used **solely as a proxy** to assess whether the synthesized fMRI preserves meaningful high-level representations. In fact, due to **its own semantic decoding limitations**, the reported semantic-level scores may **underestimate** the true alignment between the synthesized fMRI and the corresponding visual content.
>
> [1] Scotti P S, Tripathy M, Villanueva C K T, et al. MindEye2: Shared-Subject Models Enable fMRI-To-Image With 1 Hour of Data. ICML 2024.
>
> We thank the reviewer again for the valuable feedback. We hope our response adequately addresses your comments. If so, we hope you could consider a more positive evaluation of our work. If there are any remaining concerns, we would be happy to further address them during the discussion phase.

---

### Note · Authors · 2025-08-16

Dear AC and Reviewers,

We sincerely appreciate the reviewers’ thorough evaluation and constructive comments, and are especially encouraged by their acknowledgment of the strengths and contributions of our work:

* Novelty: "**addresses a fundamental challenge in computational neuroscience**" & "**biologically motivated and theoretically sound**"  & "**novel and well-motivated**" (fiTc); "**addresses high variability across humans**" (45Ff); "**improve existing decoding models as a data augmentation**" (sdRq); "**shows good potential**" & "**biologically plausible and methodologically sound**"  (DDYY)).

* Experiment: "**provide insights into neural variability patterns**" & "**experimental design is comprehensive**" & "**architecture design is well-motivated**" (fiTc); "**accounting for variance across physiological and trial-level constraints**" (45Ff); "**generates more coherent data**" (sdRq); "**state-of-the-art performance**" & "**biologically interpretable**"  (DDYY).

We are pleased to report that most of the comments were **effectively addressed** through our detailed responses and additional experiments, mainly including "extra subject evaluation" (45Ff); "clarification of training objective" (sdRq); "cross-dataset evaluation, extra baselines, training-inference efficiency" (fiTc); "few-shot adaptation baselines, multi-subject pretraining and data augmentation analysis" (DDYY). This led to positive feedback from the reviewers, i.e., "**the majority of my concerns addressed**" & "**represents a high-impact improvement to the paradigm and performance**" (fiTc); "**have answered most of my questions and concerns**" (sdRq); "**clarifications were clear and addressed all of my concerns**" (DDYY)).

In Summary:
* SynBrain presents a generative framework for visual-to-fMRI synthesis, modeling neural responses as semantic-conditioned probability distributions to capture inherent biological variability.
* SynBrain achieves state-of-the-art subject-specific and few-shot adaptation performance, providing high-quality synthetic signals to augment fMRI-to-image decoding in data-scarce settings.
* SynBrain preserves cross-trial and cross-subject functional consistency with interpretable cortical patterns aligned with neural dynamics.

We genuinely appreciate the suggestions, and believe our paper will be improved with your feedback! The additional experiments and clarifications will be reflected in the final version as well.

Sincerely,

The authors

---

### Decision · Program_Chairs · 2025-09-17

**Decision:**

Accept (poster)

**Comment:**

This paper introduces a generative model for neural responses called "SynBrain".  The model consists of a VAE and a semantic-to-neural mapping, which allows it to synthesizes high-quality fMRI signals in response to semantic inputs, and achieves high performance using few-shot learning on novel subjects. The problem setting is important and the results are impressive. The reviewers were unanimous in their assessment that the paper makes a worthwhile contribution to the literature and should be accepted.  Congratulations! Please be sure to address all reviewer comments and criticisms in the final manuscript.